# A MAX-DOAS aerosol profile retrieval algorithm for high altitude measurements: application to measurements at Schneefernerhaus (UFS), Germany

Zhuoru Wang[1, 2], Ka Lok Chan[1], Klaus-Peter Heue[1], Adrian Doicu[1], Thomas Wagner[3], Robert Holla[4], and Matthias Wiegner[5]

[1]Remote Sensing Technology Institute, German Aerospace Center (DLR), Oberpfaffenhofen, Germany
[2]Faculty of Civil, Geo and Environmental Engineering, Technical University of Munich (TUM), Munich, Germany
[3]Max Planck Institute for Chemistry, Mainz, Germany
[4]German Meteorological Office (DWD), Hohenpeißenberg, Germany
[5]Meteorological Institute, Ludwig Maximilian University of Munich (LMU), Munich, Germany

**Correspondence:** Ka Lok Chan (ka.chan@dlr.de)

**Abstract.** We present a new aerosol extinction profile retrieval algorithm for Multi-AXis Differential Optical Absorption Spectroscopy (MAX-DOAS) measurements at high altitude sites. The algorithm is based on the look-up table method. It is applied to retrieve aerosol extinction profiles from the long-term MAX-DOAS measurements (February 2012 to February 2016) at the Environmental Research Station Schneefernerhaus (UFS), Germany, (47.417°N, 10.980°E) which is located near the summit of Zugspitze at an altitude of 2,650 m. The look-up table consists of simulated $O_4$ differential slant column densities (DSCDs) corresponding to numerous possible aerosol extinction profiles. The sensitivities of $O_4$ absorption to several parameters were investigated for the design and parameterization of the look-up table. In the retrieval, simulated $O_4$ DSCDs for each possible profile are derived by interpolating the look-up table to the observation geometries. The cost functions are calculated for each aerosol profile in the look-up table based on the simulated $O_4$ DSCDs, the $O_4$ DSCD observations as well as the measurement uncertainties. Valid profiles are selected from all the possible profiles according to the cost function, and the optimal solution is defined as the weighted mean of all the valid profiles. A comprehensive error analysis is performed to better estimate the total uncertainty. Based on the assumption that the look-up table covers all possible profiles under clear sky conditions, we determined a set of $O_4$ DSCD scaling factors for different elevation angles and wavelengths. The profiles retrieved from synthetic measurement data can reproduce the synthetic profile. The result also shows that the retrieval is insensitive to measurement noise, indicating the retrieval is robust and stable. The aerosol optical depths (AODs) retrieved from the long-term measurements are compared to coinciding and co-located sun photometer observations. High correlation coefficients ($R$) of 0.733 and 0.798 are found for measurements at 360 and 477 nm, respectively. However, especially in summer the sun photometer AODs are systematically higher than the MAX-DOAS retrievals by a factor of ~2. The discrepancy might be related to the limited measurement range of the MAX-DOAS, and is probably also related to the decreased sensitivity of the MAX-DOAS measurements at higher altitudes. The MAX-DOAS measurements indicate the aerosol extinction decreases with increasing altitude during all seasons, which agrees with the co-located ceilometer measurements. Our results also show maximum AOD and

maximum Ångström exponent in summer which is consistent with observations from an AERONET station located ∼43 km of the MAX-DOAS.

## 1 Introduction

Atmospheric aerosols play an important role in the atmospheric physics and chemistry. They affect the atmospheric radiation budget by absorbing and scattering radiation, as well as providing nuclei for the formation of clouds (Haywood and Boucher, 2000; Bellouin et al., 2005; Li and Kou, 2011; Heald et al., 2014). Aerosols also have significant impacts on global climate change, local air quality and visibility (Bäumer et al., 2008; Levy II et al., 2013; Viana et al., 2014). Moreover, exposure to atmospheric aerosols can be harmful to human health (Valavanidis et al., 2008; Brook et al., 2010; Karanasiou et al., 2012). Besides primary aerosols which are directly introduced into the atmosphere, aerosols can also be secondarily formed through chemical reactions (Hinds, 2012). A significant increasing amount of anthropogenic aerosols and precursors have been released into the atmosphere since the industrial revolution (Liu et al., 1991; Junker and Liousse, 2008) which becomes a widely concerned environmental problem in recent years. Aerosols can be long-range transported and hence influence regions far from the sources (Wiegner et al., 2011; Almeida-Silva et al., 2013; Lee et al., 2013; Zhang et al., 2014; Chan and Chan, 2017; Chan, 2017; Chan et al., 2018). The properties and vertical distribution of aerosols vary strongly with time and location. Therefore, it is important to measure the spatial and temporal variations of aerosols for the better understanding of the role of aerosols in atmospheric processes. In addition, anthropogenic contribution to atmospheric aerosol load is one of the largest uncertainties in climate forcing assessments. Accurate measurements of aerosol optical properties are necessary for the further assessment of environmental and radiative effects of aerosols (Stocker et al., 2013).

Methodologies for aerosol monitoring are mature and well established: the backbone is certainly the AERONET network of sun photometers (Holben et al., 1998) providing e.g. the spectral aerosol optical depth (AOD) from direct sun observations. They might be complemented by active lidar remote sensing to provide range-resolved information. The latter includes research lidars (e.g., Pappalardo et al., 2014) and networks of ceilometers (e.g., Wiegner et al., 2014; Cazorla et al., 2017). These measurements provide – depending on the complexity of the system – the vertical distribution of the particle backscatter and extinction coefficient at typically one to three wavelengths with a very high vertical resolution in the order of 10 m, however, the uncertainty of the retrieved AOD is larger than that of sun photometers due to the restrictions of the measurement range. In the case of ceilometers inherent assumptions of the data evaluation further add to the uncertainty. Recently, the potential of Multi-AXis Differential Optical Absorption Spectroscopy (MAX-DOAS) for range-resolved aerosol retrievals was investigated as well (Platt and Stutz, 2008; Wagner et al., 2004; Frieß et al., 2006).

Ground-based MAX-DOAS is a remote sensing technique for measuring atmospheric aerosols and trace gases. MAX-DOAS instruments measure the spectra of scattered sunlight at several different viewing directions, and information of trace gas absorption along the light paths can be obtained by applying the differential optical absorption spectroscopy (DOAS) method to the ultraviolet-visible (UV-VIS) band. The retrieval of aerosol extinction profiles from MAX-DOAS measurements typically relies on the absorption signal of oxygen collision complex ($O_4$). As the vertical distribution profile of $O_4$ is well-known

and stable, it is an ideal indicator of the atmospheric distribution of photon paths. Photon paths of scattered sunlight can be influenced by aerosols and hence change the measured $O_4$ slant columns. Therefore, aerosol vertical extinction profiles can be retrieved by fitting the $O_4$ observations to radiative transfer simulations. Since the experimental setup is relative simple and inexpensive, MAX-DOAS instruments have been widely used to measure the vertical distribution of atmospheric aerosols and

trace gases in the past two decades (e.g., Hönninger et al., 2004; Irie et al., 2008; Li et al., 2010; Clémer et al., 2010; Frieß et al., 2011; Halla et al., 2011; Irie et al., 2011; Vlemmix et al., 2011; Wagner et al., 2011; Li et al., 2013; Ma et al., 2013; Wang et al., 2014a; Chan et al., 2015; Jin et al., 2016; Wang et al., 2016; Chan et al., 2017).

In the retrieval of vertical profile information from MAX-DOAS measurements, the aerosol profile is usually regarded as the state vector ($x$) and the measured $O_4$ differential slant column densities (DSCDs) of each scanning cycle are regarded as

the measurement vector ($y$). The radiative transfer model used to simulate the $O_4$ DSCDs is regarded as the forward model ($F$). As the radiative transfer in the atmosphere is non-linear, the retrieval is a non-linear problem. Moreover, the retrieval is ill-posed, which means the information contained in the observation is insufficient to determine a unique solution. In many of the other MAX-DOAS studies (e.g., Frieß et al., 2006; Clémer et al., 2010; Frieß et al., 2011; Irie et al., 2011; Wang et al., 2014a, 2016; Chan et al., 2017), aerosol profiles are retrieved using the optimal estimation method (OEM) (Rodgers, 2000).

The inversion of the aerosol profile is solved iteratively by minimizing the cost function. Vertical profile information can also be retrieved from MAX-DOAS observations using parameterized approaches (e.g., Lee et al., 2009; Li et al., 2010; Vlemmix et al., 2011; Wagner et al., 2011; Sinreich et al., 2013). These methods simplifies aerosol profiles as limited parameters, e.g., aerosol optical depth (AOD), layer height, shape parameter and etc. (Wagner et al., 2011; Hartl and Wenig, 2013). The optimal solution is usually determined by minimizing the difference between simulations and measurements.

However, as the retrieval is ill-posed and errors exist in both measurements and simulations, the profile with the lowest cost function may not be the one closest to the true profile. Moreover, in the typical OEM-based algorithms, the iteration stops as soon as the cost function is smaller than a certain threshold. Therefore, the retrieved profile is not necessarily the one with the smallest cost function. At high altitude sites, the aerosol profile retrieval is more challenging, as the $O_4$ concentration and the aerosol load are both much lower than that at low altitude sites. The vertical gradient of the aerosol extinction is also

much smaller and the relative contribution from aerosols above the retrieval height to the total AOD is more significant. As a result, the signal to noise ratio (SNR) of high altitude MAX-DOAS measurements is much lower and hence affects the retrieval quality.

In this paper, we present a new MAX-DOAS aerosol profile retrieval algorithm suitable for high altitude measurements. It is based on an $O_4$ DSCD look-up table. The look-up table includes simulated $O_4$ DSCDs corresponding to a very large number

of aerosol extinction profiles. Our retrieval algorithm is applied to MAX-DOAS observations at the Environmental Research Station Schneefernerhaus (Umweltforschungsstation Schneefernerhaus, UFS). The UFS is located close to the summit of Zugspitze (2,962 m above sea level), the highest mountain of Germany, at an altitude of 2,650 m. The $O_4$ concentration at Zugspitze is ∼40% lower compared to sea level. As the measurement site is surrounded by the mountainous area of the Alps and far from polluted areas, the aerosol load is much lower than at low altitude sites. The annual averaged AOD measured by

the sun photometer at the UFS is around 0.1 at $350 - 500$ nm. Moreover, the surface around the UFS is very complex which

complicates the radiative transfer simulation. As a result, the model errors are larger compared to the flat and simple surfaces. In the study, we first analyzed the simulation uncertainty caused by the simplification of topography definition (see Section 3.3). Then we studied the sensitivity of $O_4$ absorption to several parameters (see Section 3.4 and Appendix B). Based on the results, we designed the $O_4$ DSCD look-up table and the inversion method (see Sections 3.5 to 3.8). In Section 3.9, we present our method for determining the $O_4$ DSCD scaling factors based on the look-up table. Discussions of the retrieved aerosol profiles from the long-term measurements at the UFS are presented in Section 4.

## 2 Measurements

### 2.1 MAX-DOAS measurements

The MAX-DOAS instrument is set up on the platform on the 5[th] floor of the UFS (47.417°N, 10.980°E), about 20 m above ground level which is about 2,650 m about sea level. The instrument consists of a scanning telescope, a stepping motor which controls the viewing zenith angle of the telescope, as well as two spectrometers covering ultraviolet (UV) and visible (VIS) wavelength bands. Incoming sunlight is redirected by a prism reflector and a quartz fiber bundle to the spectrometers for spectral analysis. The field of view (FOV) of the instrument is about 0.95°. Two spectrometers (OMT Instruments, OMT ctf-60) each equipped with a CCD detector are used to measure the spectra of both UV ($320 - 478$ nm) and VIS ($427 - 649$ nm) wavelength ranges. The full width half maximum (FWHM) spectral resolutions of the UV and VIS spectrometers are about 1.0 and 0.6 nm, respectively. The scanning direction of the telescope is controlled by the stepping motor.

As the measurement geometry is limited by the topography, the viewing azimuth angle of the telescope was adjusted to the due south (180°) with the lowest elevation angle of 1°. Each scanning cycle consists of measurements at elevation angles ($\alpha$) of 90° (zenith), 30°, 20°, 10°, 5°, 2° and 1°. A single measurement at each elevation angle lasts for $\sim$1 min, and a full scanning cycle takes about 10 min. The recorded spectrum of each measurement is the sum of the CCD readouts within $\sim$1 min. In order to optimize the measurement SNR, avoid saturation and achieve a constant signal level, the data acquisition software automatically adjusts the exposure time of each readout to make the maximum count close to 70% of saturation level (65,535 counts). Depending on the intensity of received light, the exposure time of each readout varies from tens of milliseconds to a few seconds. The measurements of UV and VIS bands are taken by the two spectrometers simultaneously, but their exposure times are adjusted individually. The instrument takes measurements continuously during daytime (solar zenith angle (SZA) $< 85°$), but during the noon (175° $<$ solar azimuth angle (SAA) $<$ 185°) and twilight periods (85° $<$ SZA $<$ 92°), the instrument takes only zenith measurements.

The MAX-DOAS instrument is running since February 2012. However, the measurement was interrupted between February 2013 and July 2013 due to instrument maintenance. In February 2016, the measurement was interrupted again and the VIS spectrometer was found to be degraded. In this paper, we present four years of MAX-DOAS measurements from February 2012 to February 2016.

## 2.2 Sun photometer measurements

Next to the MAX-DOAS instrument, a sun photometer is installed at the UFS, which provides measurements of radiances at 12 wavelengths between 340 and 1640 nm with a temporal resolution of 1 s. The instrument was developed at the Meteorological Institute of Ludwig Maximilian University of Munich (LMU) based on a system operated in the framework of the SAMUM campaigns (Toledano et al., 2009, 2011) but with improved electronics and data acquisition developed by Physikalische Messsysteme Ltd. In this study, the AODs derived from sun photometer measurements applying the well-established Rayleigh calibration method were used for the inter-comparison with the MAX-DOAS retrieval. For this purpose, AOD measurements at 340 and 380 nm were interpolated to 360 nm while AODs at 477 nm were interpolated from the measurements at 440 and 500 nm. The interpolation followed the Ångström exponent method. Measurements were given as hourly averages. Due to the reduced accuracy under large SZA, only the measurements between 10:00 and 14:00 UTC each day were used. In order to ensure the data quality, only cloud-free conditions and periods of stable aerosol abundance (variability of radiances below 5% within one hour) were considered. These requirements reduce the number of available sun photometer measurements considerably. Note that the AOD is often below 0.02 at the relevant wavelengths with an uncertainty in the order of $\pm 0.015$ due to calibration errors, Rayleigh correction and radiometric accuracy. As the uncertainty of the AOD measured by the sun photometer is relatively large, the uncertainty of the Ångström exponent would be further amplified. Consequently they are not used in this study.

Aerosol optical properties not available from UFS-measurements but required for our MAX-DOAS inversion scheme (single scattering albedo and phase function) were estimated from the AERONET measurements at Hohenpeißenberg, which is located at an altitude of 980 m and approximately 43 km north of the UFS. The AERONET data were available at 440, 675, 870 and 1,020 nm, therefore, the data at 360 nm were extrapolated, and the data at 477 nm were interpolated. As Hohenpeißenberg and UFS are located at different altitudes, the aerosol optical properties might be slightly different. Therefore, we have analyzed the uncertainties caused by the differences in single scattering albedo and phase functions through a sensitivity analysis. The result shows that the influences of aerosol optical properties are in general less than 3%, see Appendix B3. Some other MAX-DOAS studies also found that aerosol optical properties show only small impacts on aerosol profile retrieval (e.g., Chan et al., 2019).

## 2.3 Ceilometer measurements

The UFS is also equipped with a Lufft (previously Jenoptik) ceilometer (model: CHM15kx, see Wiegner and Geiß (2012)) operated by the German Weather Service (DWD). Ceilometers are single-wavelength backscatter lidars, and the received signals follow the well-known lidar equation (Wiegner et al., 2014). The CHM15kx is eye-safe and fully automated which allows unattended 24/7 operation. It can be used to monitor aerosol layers (e.g., volcanic ash, see Schäfer et al. (2011)), validating meteorological and chemistry transport models (see, e.g., Emeis et al. (2011)), and is foreseen for model assimilation (e.g., Wang et al., 2014b; Warren et al., 2018; Chan et al., 2018).

The CHM15kx ceilometer is equipped with a diode-pumped Nd:YAG laser emitting laser pulses at 1,064 nm. The received backscatter signals are stored in 1,024 range bins with a resolution of 15 m. The temporal resolution is set to 15 s. The signals are corrected for incomplete overlap by a correction function provided by the manufacturer.

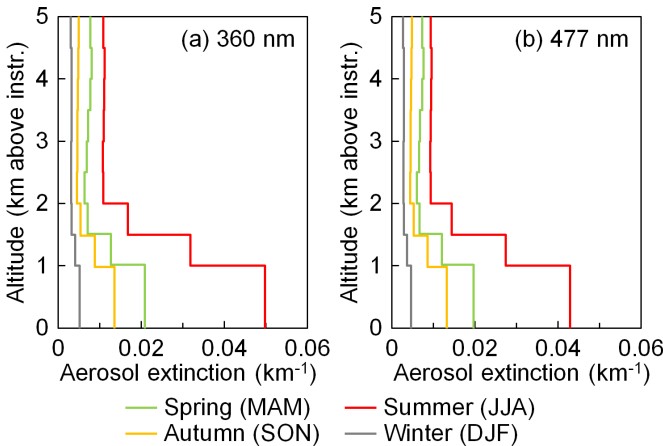

**Figure 1.** Seasonal average aerosol extinction profiles extracted from ceilometer measurements.

A strict retrieval of the particle extinction coefficient from ceilometer measurements is not possible due to the unknown lidar ratio; furthermore, exploitation of the signal in the range of incomplete overlap is subject to errors. Thus, in order to convert the ceilometer measurements to aerosol extinction profiles, we followed an approach mentioned in Wagner et al. (2019). The range corrected attenuated backscatter data from July 2016 to December 2017 were seasonally averaged. Data of the altitude between 500 m and 5 km above instrument were averaged with a vertical grid resolution of 500 m. Data below 500 m were assumed to be constant, following the values at 500 m. The extinction coefficients were first calculated by scaling the attenuated backscatter profiles ($\beta^*$) to the seasonal average AODs at 360 and 477 nm obtained from the sun photometer. The extinction profiles were then used to correct for the attenuation of the backscatter profiles following the lidar equation (Klett, 1981; Fernald, 1984). The corrected backscatter profiles ($\beta$) were then scaled to the AODs at 360 and 477 nm measured by the sun photometer to obtain the extinction profiles, see Fig. 1. Note that the ceilometer measures at 1,064 nm and the optical properties of aerosols depend on the wavelength. Therefore, the uncertainties of these profiles are very large and they should be considered as qualitative only.

The results shown in Fig. 1 indicate that the aerosol load at the UFS is highest in summer (June, July and August) and lowest in winter (December, January and February). The seasonal results also indicate large variations of the aerosol load from the surface up to 2 km. Above 2 km the variability is smaller, however, their contribution to the total column is still substantial ($\sim$30 – 50%).

## 3 Aerosol profile retrieval method

In this study, we developed an aerosol profile retrieval algorithm for MAX-DOAS measurements based on the look-up table method. According to the measurement sensitivity, we first parameterized the aerosol profile as the aerosol extinction coefficients of three altitude layers and defined a profile set which is assumed to include all possible profiles. $O_4$ DSCDs corresponding to each profile in the set were simulated and stored in the look-up table. In the retrieval, $O_4$ DSCDs are calculated from the measured spectra and then compared to the simulated ones corresponding to each profile of the set using a cost function. According to the cost function, valid profiles are selected from the set, and the optimal solution is defined as the weighted mean of all the valid profiles.

### 3.1 $O_4$ DSCD calculation

The DSCDs of $O_4$ were derived from both UV and VIS spectra using the DOAS technique (Platt et al., 1979; Platt and Stutz, 2008). In the retrieval, DSCD is defined as the difference between the slant column density (SCD) of each off-zenith spectrum ($\alpha \neq 90°$) and the corresponding zenith reference spectrum ($\alpha = 90°$). The QDOAS spectrum analysis software (version 3.2) developed by BIRA-IASB (http://uv-vis.aeronomie.be/software/QDOAS/) was used for the spectral fitting analysis. The calibration of the spectrometers was performed by fitting the measured solar spectra to the literature solar reference (Chance and Kurucz, 2010). All the measured spectra were first corrected for offset and dark current.

Details of the DOAS fit settings for both bands are listed in Table 1. The fitting windows were determined according to both the absorption signal of $O_4$ and the SNR of the spectrometers. For UV spectra, the fitting window is $338 - 370$ nm, which is the same as most of the other MAX-DOAS studies (e.g., Clémer et al., 2010; Wang et al., 2014a; Kreher et al., 2019), and it covers the strong absorption peak at 360.8 nm and a weak absorption peak at 344 nm. For VIS spectra, because the spectral range of the spectrometer begins at 427 nm and the SNR close to the spectral edges is low, we therefore adapted a smaller fitting window of $440 - 490$ nm, which is a bit narrower than the fitting window of $425 - 490$ nm commonly used in other MAX-DOAS studies (e.g., Clémer et al., 2010; Chan et al., 2017; Kreher et al., 2019). The VIS fitting window covers the strong absorption peak at 477 nm and a weak absorption peak at 446.5 nm. As the temperature at the UFS typically varies between 263 K and 279 K (Risius et al., 2015), trace gas absorption cross sections measured at 273 K were used in the DOAS fit. Absorption cross sections of several trace gases as well as a synthetic ring spectrum were included in the DOAS fit. For each scanning cycle, the zenith spectra before and after the cycle were temporally interpolated to the measurement time of each off-zenith spectrum. The broad band spectral structures caused by Rayleigh and Mie scattering were removed by including a low order polynomial in the DOAS fit. Small shift and squeeze of the wavelengths were allowed in the wavelength mapping process in order to compensate small uncertainties caused by the instability of the spectrograph.

The root mean square (RMS) of fit residual was used to evaluate the performance of the DOAS fit. DSCDs with residual RMS larger than $1 \times 10^{-3}$ were not considered in the following analysis. Under cloud-free condition, the residual RMS of most of the UV spectra varies between $5 \times 10^{-4}$ and $9 \times 10^{-4}$, while the residual RMS of most of the VIS spectra varies between

$2 \times 10^{-4}$ and $5 \times 10^{-4}$. This is because both the light intensity and the $O_4$ absorption are stronger at the VIS band, hence the measurement SNR is higher.

**Table 1.** The DOAS fit settings for UV $(338 - 370\,\text{nm})$ and VIS $(440 - 490\,\text{nm})$ bands.

| Species | Temperature | Fitting window | | Reference |
| --- | --- | --- | --- | --- |
| | | $338 - 370\,\text{nm}$ (UV) | $440 - 490\,\text{nm}$ (VIS) | |
| CHOCHO | 296 K | | ✓ | Volkamer et al. (2005) |
| HCHO | 273 K | ✓ | | Chance and Orphal (2011) |
| $H_2O$ | 296 K | ✓ | ✓ | HITEMP 2010, Rothman et al. (2010) |
| $NO_2$[a] | 273 K | ✓ | ✓ | Bogumil et al. (2003) |
| $NO_2$[a] | 220 K | ✓ | ✓ | Bogumil et al. (2003) |
| $O_3$[b] | 273 K | ✓ | ✓ | Serdyuchenko et al. (2014) |
| $O_3$[b] | 223 K | ✓ | ✓ | Serdyuchenko et al. (2014) |
| $O_4$ | 293 K | ✓ | ✓ | Thalman and Volkamer (2013) |
| Ring | | ✓ | ✓ | Chance and Spurr (1997) |
| Polynomial | | $5^{\text{th}}$ order | $5^{\text{th}}$ order | |
| Intensity offset | | linear | linear | |

[a] $I_0$ correction is applied with SCD of $10^{17}$ molec/cm$^2$ (Aliwell et al., 2002).

[b] $I_0$ correction is applied with SCD of $10^{20}$ molec/cm$^2$ (Aliwell et al., 2002).

## 3.2 Cloud screening

The aerosol profile retrieval requires the forward simulation of the radiative transfer in the atmosphere. As the radiative transfer is rather complicated for cloudy sky condition, the forward simulation usually assumes a cloud-free atmosphere. The aerosol retrieval might result in large uncertainty under cloudy or foggy conditions. Therefore, it is important to filter out the measurements taken under cloudy or foggy conditions. In this study, a colour index (CI) (Wagner et al., 2014, 2016) based cloud screening approach was applied to filter out cloudy measurements. The CI is defined as the ratio of radiative intensities at 330 and 390 nm in this study. Larger CI indicates the UV/VIS intensity ratio is higher, hence, the sky is more blue. Our cloud screening method is presented in Appendix A. The cloud screening results during the entire measurement period are summarized in Table 2. Among the four seasons, the percentage of cloudy measurements is highest in summer and lowest in winter. In total, about 60% of the zenith measurements were determined as cloudy scenes, and the corresponding scanning cycles were not used in the aerosol profile retrieval.

## 3.3 Topography effect and the simplification in radiative transfer model

The topography around the UFS is quite complex, which complicates the radiative transfer simulations. As shown in Fig. 2, the surface altitude varies between 600 and 2,800 m a.s.l. along the viewing direction of the MAX-DOAS instrument. Fig. 2

**Table 2.** Summary of cloud screening results.

| Season | Number of measurements | Number of cloudy measurements | Percentage of cloudy measurements |
|---|---|---|---|
| Spring (Mar, Apr, May) | 17,728 | 10,677 | 60.2% |
| Summer (Jun, Jul, Aug) | 21,360 | 14,259 | 66.8% |
| Autumn (Sep, Oct, Nov) | 24,259 | 13,519 | 55.8% |
| Winter (Dec, Jan, Feb) | 17,007 | 9,264 | 54.5% |
| Annual | 80,354 | 47,719 | 59.4% |

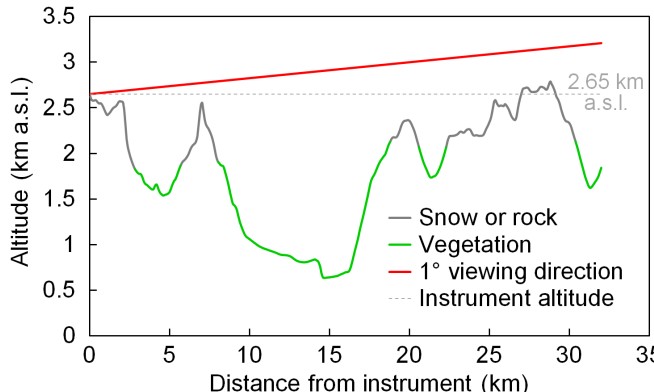

**Figure 2.** Altitude and type of the ground surface under the viewing direction (due south) of the MAX-DOAS at the UFS. Both the altitude data and surface type are obtained from Google Earth.

also shows the type of surface in different colours which includes forests, meadows, rocks, etc. Some parts of the surface are seasonally or permanently covered by snow, while some steep slopes cannot be covered by snow even in winter.

Three-dimensional radiative transfer models (RTMs) can consider such a complex terrain, but they are computational expensive and unaffordable for retrieval. Due to the limitation of the two-dimensional RTM LIDORT (Spurr et al., 2001; Spurr, 2008) used in the study, we simplified the ground topography to a flat surface at an altitude of 2,650 m a.s.l in the radiative transfer simulations. In order to estimate the error caused by this simplification, we investigated using the three-dimensional RTM TRACY-2.

TRACY-2 is a full spherical Monte-Carlo atmospheric RTM (Deutschmann, 2008; Wagner et al., 2007), which allows to simulate three-dimensional radiative transport as well as two-dimensional variation of the surface height. The model was compared to other RTMs and very good agreement was found (Wagner et al., 2007). We also did an inter-comparison with LIDORT. The result shows that with the same definition of topography and atmosphere, the difference between the $O_4$ DSCDs simulated by the two RTMs is less than 3%.

For the three-dimensional simulations carried out in this study, a pseudo-reality topography was defined with the exact ground altitude (obtained from Google Earth) in the azimuth direction of the MAX-DOAS measurements taken into account, whereas in the dimension orthogonal to this direction, the surface altitude was set constant. This simplification was chosen to reduce the computational effort. We feel that this approach is justified since the atmospheric light paths in the viewing direction of the instruments can be very large (up to several tens of kilometers), it is most important to take this variation of the surface altitude along this direction into account, whereas the influence of the orography perpendicular to this direction is expected to be small.

Simulations were performed with all the combinations of three different SZAs (30°, 50° and 70°), three different relative solar azimuth angles (RAAs) (30°, 60° and 90°) and two different aerosol extinction profiles (an aerosol-free profile and a box-shape profile with AOD = 0.12 and box height = 3 km), i.e. altogether 18 cases. For each case, $O_4$ DSCDs at 360 and 477 nm were simulated with both the flat surface at 2,650 m and the pseudo-reality topography using TRACY-2. The relative errors of $O_4$ DSCDs simulated with the flat surface compared to those simulated with the pseudo-reality topography are calculated. A fixed surface albedo of 0.07 was used in the simulations. For both wavelengths, the single scattering albedo was set to 0.93 and the phase function was defined as a Henyey-Greenstein phase function with the asymmetry parameter set to 0.68. The atmospheric profile was defined as the US standard mid-latitude atmosphere (Anderson et al., 1986). Fig. 3 shows the results of some of the cases: (a) and (b) show the results of six cases with SZA = 50° and different RAAs and both aerosol extinction profiles; (c) and (d) show the results of six cases with RAA = 60° and different SZAs and also both aerosol extinction profiles.

**Table 3.** Systematic and random errors caused by the topography simplification. Results are calculated from the relative differences of $O_4$ DSCDs simulated with a flat surface at 2,650 m comparing to those simulated with the pseudo-reality surface in 18 cases (see text). The mean of the relative difference for each elevation angle and each wavelength is considered as the systematic error. The standard deviation of the relative difference is considered as the random error.

| Elevation angle | UV (360 nm) | | VIS (477 nm) | |
|---|---|---|---|---|
| | Systematic error (%) | Random error (%) | Systematic error (%) | Random error (%) |
| 1° | -3.19 | 1.99 | -2.30 | 2.24 |
| 2° | -3.69 | 1.64 | -1.90 | 2.21 |
| 5° | -3.42 | 1.60 | -2.48 | 1.57 |
| 10° | -4.12 | 2.32 | -3.51 | 2.24 |
| 20° | -4.74 | 3.09 | -3.93 | 4.63 |
| 30° | -5.08 | 5.44 | -3.91 | 5.84 |

As shown in all the panels of Fig. 3 as well as in all the other cases not shown, $O_4$ DSCDs simulated with the flat surface are in general slightly underestimated compared to the pseudo-reality topography. The difference could be explained by the scattering in the valleys where the concentration of $O_4$ is higher. For the flat surface at 2,650 m, the light paths below 2,650 m would not be taken into account, and hence the $O_4$ DSCDs would be underestimated. Moreover, the relative error has no obvious correlation with elevation angle, SZA, RAA and aerosol load. This is because the light path below 2,650 m is influenced by

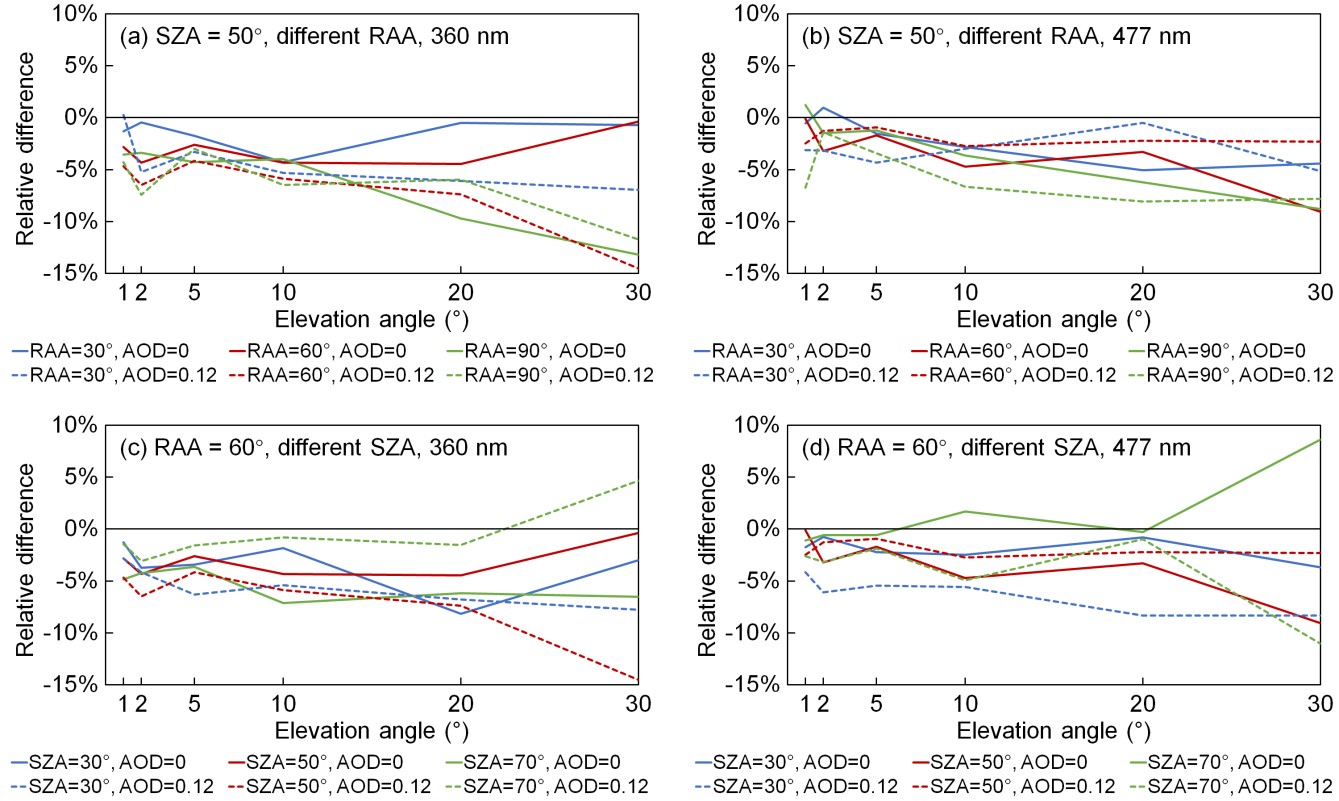

**Figure 3.** Relative differences of $O_4$ DSCDs at (a, c) 360 nm and (b, d) 477 nm simulated with a flat surface at 2,650 m comparing to the $O_4$ DSCDs simulated with the pseudo-reality topography. (a) and (b) show the results simulated with the same SZA of 50° and different RAAs (relative solar azimuth angles) of 30°, 60° and 90°; (c) and (d) show the results simulated with the same RAA of 60° and different SZAs of 30°, 50° and 70°. Solid curves are the results simulated under aerosol-free condition, and dashed curves are the results simulated with a box-shape profile with AOD = 0.12 and box height = 3 km.

the topography, and the influence differs with the observation geometry. In addition, the light path is also influenced by the aerosols both below and above 2,650 m. Concerning the fact that only a pseudo-reality surface and a constant surface albedo is used in the study, the actual error caused by the topography simplification is expected to be much more complicated.

In order to make the compensation feasible, we consider the error as the combination of a systematic error and a random

5  error. Based on the results of all the 18 cases of this study, the mean bias for each elevation angle and each wavelength is considered as the systematic error, while the standard deviation of the relative difference is considered as the random error, see Table 3. In the aerosol profile retrieval, systematic errors are first corrected from the measured $O_4$ DSCDs, while random errors are included in the error budget in the calculation of cost functions (see Section 3.7.2). In the following text, measured $O_4$ DSCDs refer to the values corrected by the systematic error unless otherwise mentioned.

### 3.4 Sensitivity analysis

In order to make full use of the measurement sensitivity and reduce unnecessary computational efforts, our retrieval algorithm was designed according to the sensitivity of $O_4$ absorption. We performed several sensitivity analyses to determine the optimal vertical grid, step size of the aerosol extinction for each layer and the maximum aerosol extinction. In addition, these sensitivity analyses also help to estimate the measurement and model errors which are very important for the retrieval. The sensitivity analyses are based on forward simulations of $O_4$ DSCDs using LIDORT. We tested the sensitivities of $O_4$ absorption to surface albedo, aerosol optical properties and aerosol vertical profile. The results of the sensitivity analyses are shown in Appendix B. The extreme and median values of the parameters are also discussed in that section.

### 3.5 Parameterization of the aerosol extinction profile

As discussed in Appendix B4, $O_4$ absorption is insensitive to aerosols above 2 km. Therefore, our retrieval only focuses on aerosols between 0 and 2 km above the MAX-DOAS instrument (i.e. $2,650 - 4,650$ m a.s.l.). In order to limit the complexity of the retrieval, avoid unreasonable results and make full use of the measurement sensitivity, we parameterize the aerosol extinction profile as aerosol extinction extinctions in three layers. The thicknesses of the two lower layers are defined as 0.5 km. Due to the lower sensitivity at high altitude, the thickness of the third layer is set to 1 km. The aerosol profile is denoted as a 3-dimensional state vector $x$,

$$
x = \begin{pmatrix} \sigma_1 \\ \sigma_2 \\ \sigma_3 \end{pmatrix}, \tag{1}
$$

where $\sigma_1$ is the aerosol extinction coefficient between 0 and 0.5 km ($2,650 - 3,150$ m a.s.l.), $\sigma_2$ is the aerosol extinction coefficient between 0.5 and 1 km ($3,150 - 3,650$ m a.s.l.), and $\sigma_3$ is the aerosol extinction coefficient between 1 and 2 km ($3,650 - 4,650$ m a.s.l.). The definition of $x$ is illustrated in Fig. 4 (a). The vertical resolution of our retrieval grid is lower comparing to many other studies (e.g., Clémer et al., 2010; Chan et al., 2017; Tirpitz et al., 2020), however, the vertical gradient of aerosol extinction at such a high altitude site is expected to be small and this is also proved by the ceilometer measurements. Therefore, the coarse resolution is considered to be sufficient for the retrieval of the UFS MAX-DOAS measurements.

In order to formulate the look-up table, we defined a profile set (denote as $X_{LUT}$) which is assumed to include all possible aerosol extinction profiles under cloud-free condition. $X_{LUT}$ is a finite set of $x$, and the variation steps of $\sigma_1$, $\sigma_2$ and $\sigma_3$ were determined according to the sensitivity and accuracy of measurement. $X_{LUT}$ includes only the profiles with reasonable shapes, and the variation range of $\sigma_1$, $\sigma_2$ and $\sigma_3$ covers the actual aerosol load at the UFS. In this way, unreasonable and unrealistic retrieval results can be avoided.

As discussed in Appendix B6, the measurement sensitivity decreases with increasing surface aerosol extinction, and the sensitivity is very low when the surface aerosol extinction coefficient exceeds $0.3\,\text{km}^{-1}$. Therefore, $\sigma_1$ is defined to vary between 0 and $0.3\,\text{km}^{-1}$. The variation step increases from $0.001\,\text{km}^{-1}$ per step to $0.02\,\text{km}^{-1}$ per step with increasing aerosol

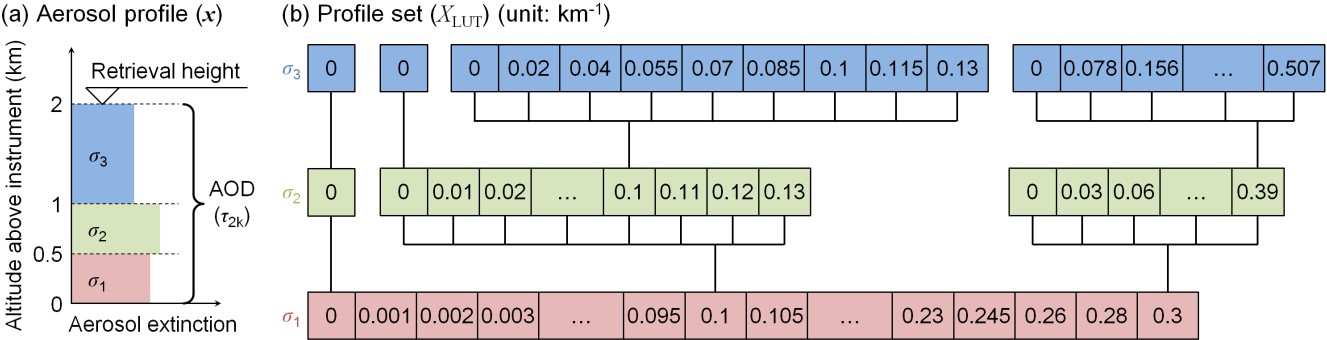

**Figure 4.** Definitions of (a) the parameterized aerosol profile ($\boldsymbol{x}$) and (b) the profile set ($X_{\mathrm{LUT}}$). Note that only some representative nodes are shown in Panel (b).

extinction, so that the difference of $O_4$ DSCD per step is similar to the average spectral fitting error ($\sim$2%). In total, we define 65 values for $\sigma_1$, see Table 4.

As illustrated in Fig. 4 (b), the values of $\sigma_2$ and $\sigma_3$ are defined as a tree, which means we define different values of $\sigma_2$ for each $\sigma_1$, and the values of $\sigma_3$ are also defined depending on $\sigma_2$. According to the ceilometer observations at the UFS, strong elevated aerosol layers are unlikely to exist under cloud-free conditions, therefore we allow only weak elevated layers in designing the profile set. We assume that for reasonable profiles, $\sigma_2$ should not exceed $\sigma_1$ by more than 30%, and $\sigma_3$ should not exceed $\sigma_2$ by more than 30%, either. According to the sensitivity, for each value of $\sigma_1$ ($\sigma_1 > 0$), we define 14 possible values for $\sigma_2$ which varies from 0 to $1.3\sigma_1$ with a step size of $0.1\sigma_1$. In case $\sigma_1 = 0$, elevated layers are not considered, then $\sigma_2$ and $\sigma_3$ can only be 0. Similarly, $\sigma_3$ varies between 0 and $1.3\sigma_2$. Due to the lower measurement sensitivity at high altitude, we define 9 possible ratios between $\sigma_3$ and $\sigma_2$ (see Table 4). In case $\sigma_2 = 0$, $\sigma_3$ can only be 0.

$X_{\mathrm{LUT}}$ includes the profiles with all the combinations of $\sigma_1$, $\frac{\sigma_2}{\sigma_1}$ and $\frac{\sigma_3}{\sigma_2}$. For each of the 64 nonzero values of $\sigma_1$, there are $1 + (13 \times 9) = 118$ corresponding profiles. For $\sigma_1 = 0$, there is only one profile with $\sigma_1 = \sigma_2 = \sigma_3 = 0$. Therefore, the profile set consists of $1 + 64 \times 118 = 7{,}553$ aerosol extinction profiles in total.

### 3.6 Definitions of other dimensions of the look-up table

The basic idea of the look-up table method is to replace the repetitive time-consuming computation by a pre-calculated array. In this study, we replace the forward simulation of $O_4$ DSCDs by a look-up table, so that all the possible aerosol extinction profiles can be considered in the retrieval of each measurement cycle with an affordable computational effort.

Besides the parameterized aerosol extinction profile $\boldsymbol{x}$, we consider another four input parameters for the forward simulation which can be described as a function,

$$\Delta S_{\mathrm{s}} = f(\boldsymbol{x}, \lambda, \alpha, \theta, \phi), \tag{2}$$

**Table 4.** Definition of input parameters of the $O_4$ DSCD look-up table.

| Parameter | Symbol | Number of grid points | Grid values |
|---|---|---|---|
| Aerosol extinction coefficient of $0-0.5\,\mathrm{km}$ above instrument ($\mathrm{km}^{-1}$) | $\sigma_1$ | 65 | 0, 0.001, 0.002, ..., 0.009, (0.001/step) <br> 0.01, 0.0115, 0.013, ..., 0.0265, (0.0015/step) <br> 0.028, 0.03, 0.032, ..., 0.038, (0.002/step) <br> 0.04, 0.0425, 0.045, 0.0475, (0.0025/step) <br> 0.05, 0.053, 0.056, ..., 0.077, (0.003/step) <br> 0.08, 0.085, 0.09, ..., 0.115, (0.005/step) <br> 0.12, 0.13, 0.14, ..., 0.19, (0.01/step) <br> 0.2, 0.215, 0.23, 0.245, (0.015/step) <br> 0.26, 0.28, 0.3 (0.02/step) |
| Aerosol extinction coefficient of $0.5-1\,\mathrm{km}$ above instrument ($\mathrm{km}^{-1}$) | $\sigma_2$ | 14 ($\sigma_1 > 0$) <br> 1 ($\sigma_1 = 0$) | 0, $0.1\sigma_1$, $0.2\sigma_1$, ..., $1.3\sigma_1$ ($0.1\sigma_1$/step) |
| Aerosol extinction coefficient of $1-2\,\mathrm{km}$ above instrument ($\mathrm{km}^{-1}$) | $\sigma_3$ | 9 ($\sigma_2 > 0$) <br> 1 ($\sigma_2 = 0$) | 0, $0.2\sigma_2$, $0.4\sigma_2$, $0.55\sigma_2$, $0.7\sigma_2$, $0.85\sigma_2$, $\sigma_2$, $1.15\sigma_2$, $1.3\sigma_2$ |
| Wavelength (nm) | $\lambda$ | 2 | 360, 477 |
| Elevation viewing angle (°) | $\alpha$ | 6 | 1, 2, 5, 10, 20, 30 |
| Solar zenith angle (SZA) (°) | $\theta$ | 63 | 24, 25, 26, ..., 86 (1/step) |
| Relative solar azimuth angle (RAA) (°) | $\phi$ | 122 | 0, 1, 2, ..., 121 (1/step) |

where $\Delta S_\mathrm{s}$ refers to the simulated $O_4$ DSCD, $\lambda$ represents the wavelength, $\alpha$ indicates the elevation angle, $\theta$ is the SZA, and $\phi$ is the RAA. All the input parameters are well known in the retrieval.

In order to formulate the look-up table, the input parameters need to be parameterized as a grid with finite nodes. As already presented in Section 3.5, the aerosol extinction profile ($x$) is parameterized as a profile set which consists of 7,553 possible profiles. As the simulated $O_4$ DSCDs are used to fit to the measured ones, only the data at 360 and 477 nm and at the six non-zenith elevation angles of the measurement cycles are included in the look-up table. SZA ($\theta$) and RAA ($\phi$) are parameterized as a grid with $1° \times 1°$ resolution. The grid includes 5,005 combinations of SZA and RAA, which can cover all possible solar positions for the daytime measurements at the UFS. When we obtain data from the look-up table, as the input SZA and RAA are not integers, the output $\Delta S_\mathrm{s}$ is interpolated from the data of the four adjacent nodes of the SZA-RAA grid. In total, the five input parameters are parameterized as a grid with $7,553 \times 2 \times 6 \times 5,005 = 453,633,180$ nodes. Details of the parameterization of the input parameters are summarized in Table 4.

As discussed in Appendix B, besides the input parameters we defined, $O_4$ DSCDs can also be affected by other parameters such as the ground albedo, aerosol optical properties and others. Since accurate measurements of these parameters are not available and their influence is relatively small, they are considered as uncertainties. In creating the look-up table, these

parameters were fixed to the median values. Details of the simulation settings are listed in Table 5. $O_4$ DSCDs corresponding to all the nodes of the look-up table were simulated using LIDORT.

As discussed in Appendix B5, the influence from the aerosols above 2 km is also considered as a kind of uncertainty and treated in a similar way as the other unknown parameters. In the simulations for creating the look-up table, the aerosol extinction coefficient between 2 and 4 km was defined as $0.5\sigma_3$, so that this so-called parameter is fixed to the 'median' value. Note that the aerosol extinction coefficient above 2 km is neither considered as a part of the retrieved profile nor counted in the retrieved AOD.

**Table 5.** Settings of fixed parameters in calculating the $O_4$ DSCD look-up table.

| Parameter | Value or definition |
|---|---|
| Topography | Flat surface at an altitude of 2,650 m a.s.l. |
| Surface albedo | 0.1 |
| Single scattering albedo (SSA) | 0.93 (360 nm) / 0.92 (477 nm) |
| Phase function | The 'median' phase function defined in Appendix B3 |
| Climatology | US standard profiles for profile, temperature and trace gas profiles |
| Aerosol extinction coefficient of $2-4$ km above instrument | 50% of the aerosol extinction coefficient of $1-2$ km above instrument (i.e. $0.5\sigma_3$) |
| Aerosol extinction coefficient above 4 km from instrument | 0 |

## 3.7 Error estimation

Most of the other MAX-DOAS studies only consider the spectral fitting error in their retrieval. However, this fitting error only contributes to a small part of the total error. In addition, the total error is not directly proportional to the spectral fitting error. As the measurement and simulation uncertainties play an important part in our inversion method, we perform a comprehensive error analysis for the MAX-DOAS measurement and radiative transfer simulation of $O_4$ DSCDs. In this study, error from seven major sources are taken into account in estimating the total uncertainty.

### 3.7.1 Error in measured $O_4$ DSCDs

Two error sources related to measured $O_4$ DSCDs are taken into account in the total uncertainty estimation, which are the DOAS fitting error ($\epsilon_{fit}$) and the error caused by temperature variation ($\epsilon_{temp}$).

$\epsilon_{fit}$ is the byproduct of the DSCD calculation, derived from the fit residual and the absorption cross section of $O_4$. It is proportional to the RMS of the fit residual. For low elevation angles (1°, 2°, 5°), the percentage of $\epsilon_{fit}$ comparing to the DSCD typically varies between 1% and 3% at the UV band and between 0.3% and 0.7% at the VIS band, which is rather small compared to other sources of error. However, for the elevation angle of 30°, as the absolute DSCD value is much smaller, the percentage of $\epsilon_{fit}$ can be up to ~25% and ~10% at the UV and VIS bands, respectively.

As discussed in Section 3.1, $O_4$ absorption cross section measured at 273 K was used in the DOAS fitting. However, the effective temperature of the MAX-DOAS measurements could be significantly different from 273 K. Previous studies show that $O_4$ absorption has a strong and systematic dependence on temperature (Thalman and Volkamer, 2013; Wagner et al., 2019). In order to estimate $\epsilon_{temp}$, we compared the $O_4$ DSCDs retrieved using the cross sections measured at 253 K and 293 K to those retrieved with the cross sections measured at 273 K. The comparison shows that the $O_4$ DSCDs are underestimated by 5.1% at the UV band and 2.5% at the VIS band when the effective temperature is 293 K. On the other hand, the $O_4$ DSCDs are overestimated by 6.9% at the UV band and 3.9% at the VIS band when the effective temperature is 253 K. These systematic errors are almost constant, regardless of the observation geometry. Between 253 and 293 K, the average variation rate of $O_4$ DSCD at UV band is 0.3% / K. This result is in general agreement with Wagner et al. (2019). They found that with the fitting window of $352 - 387$ nm, $O_4$ DSCDs retrieved using the cross section at 203 K are reported to be 30% smaller than those retrieved using the cross section at 293 K, i.e. 0.33% / K in average. Based on the fact that the temperature at the measurement site varies between ∼258 and 288 K during daytime in most cases, we estimate the $\epsilon_{temp}$ of all measurements as 4.5% and 2.4% of the $O_4$ DSCD at UV and VIS band, respectively.

### 3.7.2 Error in simulated $O_4$ DSCDs

Five error sources related to simulated $O_4$ DSCDs are taken into account in estimating the total uncertainty. They are the random error caused by the simplification of the topography definition ($\epsilon_{topo}$), the error caused by surface albedo ($\epsilon_{SA}$), the error caused by single scattering albedo ($\epsilon_{SSA}$), the error caused by phase function ($\epsilon_{PF}$) and the error caused by aerosols above retrieval height ($\epsilon_{2\text{-}4\,km}$).

As discussed in Section 3.3, the random error caused by the simplification of the topography definition ($\epsilon_{topo}$) of each elevation angle and each wavelength is derived from the standard deviation of the relative errors of the 18 cases simulated using the three-dimensional RTM TRACY-2. Values of $\epsilon_{topo}$ are listed in Table 3.

For the uncertainties from the other four sources ($\epsilon_{SA}$, $\epsilon_{SSA}$, $\epsilon_{PF}$ and $\epsilon_{2\text{-}4\,km}$), as discussed in Appendix B, they can be estimated by radiative transfer simulations. Since they differ under different observation geometries and different aerosol loads, we determine them using simple look-up tables in the retrieval. In order to simplify the error estimation process, we assume that the uncertainties from the four sources are only influenced by the AOD, while the influence from different vertical distribution of aerosols is neglected. In addition, from the $O_4$ DSCD look-up table, we found that $O_4$ DSCD at 5° is almost negatively correlated with AOD, while it is insensitive to the shape of profile. Therefore, we use the $O_4$ DSCD measured at 5° as the indicator for estimating the AOD in deriving uncertainty values from the error look-up tables.

The error look-up tables consist of the values of $\epsilon_{SA}$, $\epsilon_{SSA}$, $\epsilon_{PF}$ and $\epsilon_{2\text{-}4\,km}$ for all the combinations of SZA and RAA (with $1° \times 1°$ resolution) and 65 profiles of the $X_{LUT}$ with $\sigma_1 = \sigma_2 = \sigma_3$. The calculation of the error look-up tables was similar as the sensitivity study. In order to estimate the uncertainty caused by each parameter, $O_4$ DSCDs were simulated under both median and extreme values, while all the other parameters were fixed as the median settings listed in Table 5. The relative difference between the two simulations is treated as the uncertainty and stored in the look-up table.

As discussed in Appendix B1, the uncertainty caused by surface albedo ($\epsilon_{SA}$) was derived from the relative difference of the $O_4$ DSCDs simulated with the surface albedo set to 0.2 (extreme value) and 0.1 (median value).

As discussed in Appendix B2, in the estimation of the uncertainty caused by single scattering albedo ($\epsilon_{SSA}$), the extreme value was chosen as 0.997 for both the UV and VIS bands, while the median value was chosen as 0.92 and 0.93 for UV and

VIS bands, respectively.

As discussed in Appendix B3, from all the phase functions measured by the AERONET station in Hohenpeißenberg during the period of $2013-2014$, the phase function with which the simulated $O_4$ DSCDs at all elevation angles are closest to the median values was chosen as the so-called 'median' phase function. The phase function with which the simulated $O_4$ DSCDs are closest to the rank of 95% (i.e. $2\sigma$) was chosen as the 'extreme' phase function. $\epsilon_{PF}$ was derived from the relative difference

between $O_4$ DSCDs simulated with 'median' and 'extreme' phase functions.

As discussed in Appendix B5, the error caused by aerosols above $2\,\mathrm{km}$ ($\epsilon_{2\text{-}4\,\mathrm{km}}$) is treated similarly as $\epsilon_{SA}$, $\epsilon_{SSA}$ and $\epsilon_{PF}$ in the study. The so-called 'median' $O_4$ DSCDs were simulated with profiles with the aerosol extinction coefficient between 2 and $4\,\mathrm{km}$ equals to $0.5\sigma_3$ (50% of the aerosol extinction coefficient between 1 and $2\,\mathrm{km}$), while the 'extreme' values were simulated with the aerosol extinction coefficient between 2 and $4\,\mathrm{km}$ set equal to $\sigma_3$. $\epsilon_{2\text{-}4\,\mathrm{km}}$ was derived from the relative

difference between the 'extreme' and 'median' results.

### 3.7.3    Total uncertainty

We assume that the seven kinds of errors mentioned in Sections 3.7.1 and 3.7.2 follow the normal distribution, and the total uncertainty of each band and each elevation angle can be determined by the root mean square of the seven errors as

$$\epsilon = \sqrt{\epsilon_{\text{fit}}^2 + \epsilon_{\text{temp}}^2 + \epsilon_{\text{topo}}^2 + \epsilon_{SA}^2 + \epsilon_{SSA}^2 + \epsilon_{PF}^2 + \epsilon_{2\text{-}4\,\mathrm{km}}^2}. \tag{3}$$

Examples of the error budgets of two measurement cycles for both wavelength bands are shown in Fig. 5 and Fig. 6. The cycle shown in Fig. 5 was measured in summer under relatively high aerosol load (AOD at $440\,\mathrm{nm}$ measured by the sun photometer around the noon of that day was $\sim0.2$), while the cycle shown in Fig. 6 was measured in winter under relatively low aerosol load (AOD at $440\,\mathrm{nm}$ measured by the sun photometer around the noon of that day was $\sim0.015$). In addition, The former cycle was measured under a smaller SZA comparing to the latter one ($64°$ and $79°$, respectively), while the RAA was

much larger than the latter ($97°$ and $39°$, respectively). The results show that contributions from different error sources are quite different in different measurement cycles, at different wavelengths and at different elevation angles.

### 3.7.4    Other possible error sources

Besides the seven above-mentioned error sources, there are still some other sources of error which are difficult to be estimated and hence not included in the error estimation. For example:

a. Error in $O_4$ DSCD scaling factors: in this study, we found that an elevation dependent $O_4$ DSCD scaling factor is needed to bring measurements and modeled results into agreement. We determined the factors based on the statistical analysis of the

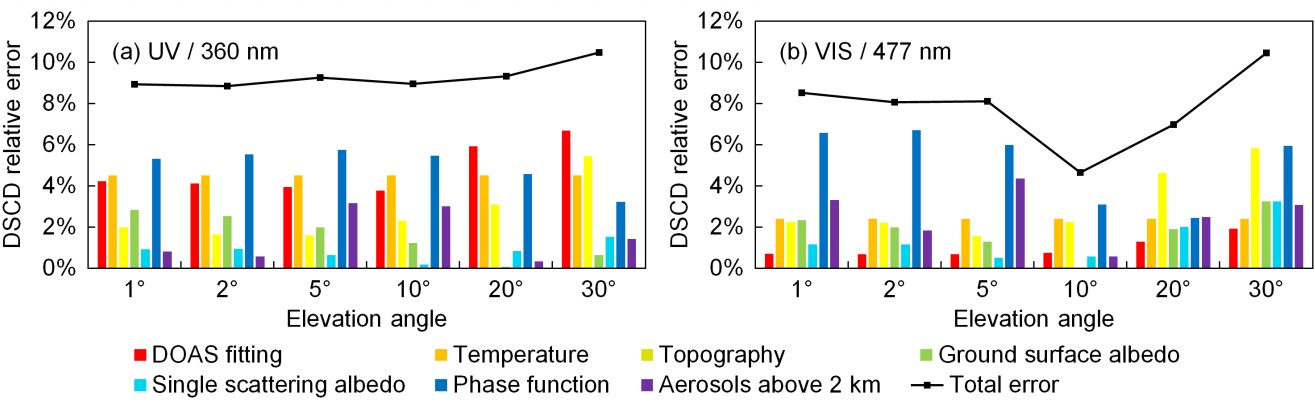

**Figure 5.** Error budget of (a) UV and (b) VIS bands of the scanning cycle on 05 Jul 2015 at ∼16:26 UTC (SZA ∼64°, RAA ∼97°). Y-axes refer to the relative error of $O_4$ DSCDs.

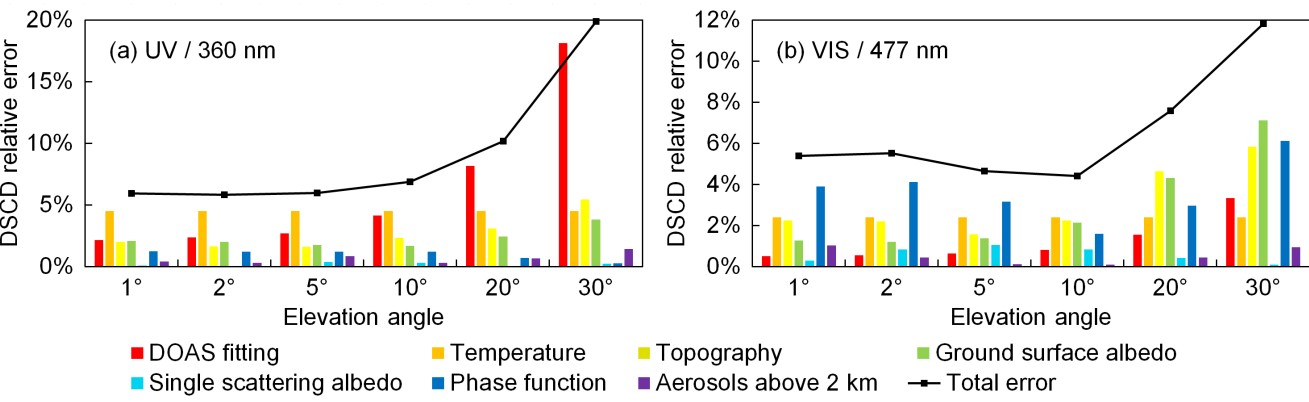

**Figure 6.** Same as Fig. 5, but for the scanning cycle on 07 Dec 2015 at ∼13:55 UTC (SZA ∼79°, RAA ∼39°).

long-term measurements, see Section 3.9. However, as it is still difficult to estimate the uncertainties of the scaling factors, they are currently not taken into account in calculating the total uncertainty.

b. Error caused by horizontal gradients of the aerosol extinction: besides its direct effect on the measurements, the complex topography might also cause systematic horizontal gradients of the aerosol extinction. For example polluted air masses from the valleys might be transported to higher altitudes according to the vertical mixing and the prevailing wind direction. Such effects can be especially important for the measurements discussed here because of the rather low AOD. Further quantification of the effects of possible horizontal gradients is beyond the scope of this study, but might be one reason for the observed elevation dependence of the $O_4$ DSCD scaling factor.

c. Error caused by the variation of atmospheric profile: the $O_4$ DSCD look-up table was calculated using the US standard climatology data, but the change of atmospheric temperature and pressure can slightly affect the $O_4$ absorption. However, since it is difficult to estimate the accurate uncertainty and real-time measurements of temperature and pressure profiles are not available, the error caused by the variation of the atmospheric profile is not taken into account in calculating the total uncertainty.

d. Systematic effect of the surface albedo on the measurements at the high altitude station: due to the dependence of the snow coverage on altitude, the surface albedo close to the instrument is typically higher than at locations far away. Since the measurements at high elevation angles are usually more sensitive to air masses closer to the instrument, they are probably stronger affected by snow and ice than measurements at low elevation angles. In this study, this effect cannot be further quantified, but it might be one reason for the need of different $O_4$ DSCD scaling factors for different elevation angles, see Section 3.9.

In order to avoid the underestimation of the measurement uncertainty, we set a relative relaxed threshold of cost functions for choosing valid profiles, see Section 3.8.

## 3.8 Inversion method

Aerosol extinction profiles are retrieved from the measured $O_4$ DSCDs of each scanning cycle. The measurements of the UV and VIS bands are retrieved separately. The measured $O_4$ DSCDs at the UV and VIS bands are fitted to the simulated $O_4$ DSCDs at 360 and 477 nm, respectively. In the retrieval, we assume the state of atmosphere being stable during a scanning cycle, and the distribution of aerosols homogeneous in horizontal direction. For a single scanning cycle, the measured $O_4$ DSCDs at the wavelength $\lambda$ are denoted as a measurement vector

$$\boldsymbol{y}_{\mathrm{m}} = \begin{pmatrix} \Delta S_{\lambda,1} \\ \Delta S_{\lambda,2} \\ \vdots \\ \Delta S_{\lambda,M} \end{pmatrix}, \tag{4}$$

where $M$ is the number of off-zenith measurements in each scanning cycle, which is 6 in this study. $\Delta S_{\lambda,1}$, $\Delta S_{\lambda,2}$, ..., $\Delta S_{\lambda,6}$ are the $O_4$ DSCDs measured at $O_4$ wavelength band $\lambda$ with the viewing elevation angles of $1°$, $2°$, $5°$, $10°$, $20°$ and $30°$, respectively.

The simulated O$_4$ DSCDs corresponding to each possible aerosol extinction profile in $X_{\text{LUT}}$ can be obtained from the look-up table. Similar to $\boldsymbol{y}_{\text{m}}$, the simulation vector $\boldsymbol{y}_{\text{s}}$ for each possible profile $\boldsymbol{x}$ is be denoted as

$$
\boldsymbol{y}_{\text{s}}(\boldsymbol{x}) = \begin{pmatrix} f(\boldsymbol{x},\lambda,\alpha_1,\theta_1,\phi_1) \\ f(\boldsymbol{x},\lambda,\alpha_2,\theta_2,\phi_2) \\ \vdots \\ f(\boldsymbol{x},\lambda,\alpha_M,\theta_M,\phi_M) \end{pmatrix}, \boldsymbol{x} \in X_{\text{LUT}}. \tag{5}
$$

Aerosol extinction profiles can be derived by fitting the forward simulation to the measured O$_4$ DSCDs. Typically, the optimal solution can be determined by minimizing the cost function, which is defined as

$$
\chi^2(\boldsymbol{x}) = [\boldsymbol{y}_{\text{m}} - \boldsymbol{y}_{\text{s}}(\boldsymbol{x})]^{\text{T}} \cdot \mathbf{S}_{\epsilon}^{-1} \cdot [\boldsymbol{y}_{\text{m}} - \boldsymbol{y}_{\text{s}}(\boldsymbol{x})], \tag{6}
$$

where $\mathbf{S}_{\epsilon}$ is the uncertainty covariance matrix. Assuming the measurements of each viewing elevation angle are independent, $\mathbf{S}_{\epsilon}$ is a diagonal matrix and its diagonal elements equal to the square of the total uncertainties of each elevation angle defined in Eq. (3),

$$
\mathbf{S}_{\epsilon} = \begin{bmatrix} \epsilon_1^2 & 0 & \dots & 0 \\ 0 & \epsilon_2^2 & \dots & 0 \\ \vdots & \vdots & \ddots & \vdots \\ 0 & 0 & \dots & \epsilon_M^2 \end{bmatrix}. \tag{7}
$$

Our cost function definition is similar to the cost functions used in many of the MAX-DOAS studies based on the OEM (e.g., Clémer et al., 2010; Frieß et al., 2016; Wang et al., 2016; Chan et al., 2017), but only includes the item related to measurement error, while the item related to the a priori profile is omitted. This is because the a priori profile is not needed in our retrieval algorithm.

$\chi^2$ indicates the difference between $\boldsymbol{y}_{\text{s}}$ and $\boldsymbol{y}_{\text{m}}$, however, as the retrieval is ill-posed and the SNR of the measurements at the UFS is low, the single profile with the lowest $\chi^2$ is not necessarily the one closest to the true profile. In order to overcome this limitation, we consider all the profiles in $X_{\text{LUT}}$ with $\chi^2(\boldsymbol{x}) \leq 1.5M$ (9 in this study) as valid profiles and calculate the weighted mean profile as the optimal result. A profile with $\chi^2 \leq M$ indicates that the measured and simulated O$_4$ DSCDs agree within the measurement errors, but in order to avoid underestimation of the measurement errors, we define the threshold as $1.5M$. The weight of each valid profile for the calculation of the optimal solution is defined as

$$
w(\boldsymbol{x}) = \frac{1/\chi^2(\boldsymbol{x})}{\sum[1/\chi^2(\boldsymbol{x})]}, \boldsymbol{x} \in X_{\text{LUT}}, \chi^2(\boldsymbol{x}) \leq 1.5M, \tag{8}
$$

and the optimal solution can be calculated as

$$\hat{\boldsymbol{x}} = \sum w(\boldsymbol{x}) \cdot \boldsymbol{x}, \boldsymbol{x} \in X_{\mathrm{LUT}}, \chi^2(\boldsymbol{x}) \leq 1.5M. \tag{9}$$

## 3.9 $O_4$ DSCD correction

Discrepancies between measured and simulated $O_4$ DSCDs are found in many other MAX-DOAS studies (Wagner et al., 2009; Clémer et al., 2010; Chan et al., 2015; Wang et al., 2016; Chan et al., 2017; Wagner et al., 2019). The discrepancies are often explained by the systematic errors of the absorption cross section of $O_4$ as well as the radiative transfer simulation, and a correction is therefore necessary. Previous studies suggested to multiply a constant scaling factor (usually between 0.75 and 0.9) to the measured $O_4$ DSCD for all elevations to correct for the systematic error (e.g., Wagner et al., 2009; Clémer et al., 2010; Chan et al., 2015; Wang et al., 2016). Some recent studies suggested elevation-dependent scaling factors. Irie et al. (2015) suggested a set of scaling factors for 477 nm which gradually decreases with increasing elevation angle, varying from 0.984 for 1° to 0.667 for 30°. Zhang et al. (2018) suggested a set of scaling factors for 360 nm which also decreases with increasing elevation angle, varying from from 1.02 for 1° to 0.909 for 30°. Chan et al. (2017) derived a set of elevation-dependent scaling factors for 477 nm by comparing modelled and measured (relative) intensity, varying from 0.792 for 1° to 0.957 for 30°. On the other hand, some other MAX-DOAS studies did not find it necessary to apply any correction to $O_4$ DSCDs. For example, Frieß et al. (2011) reported that for the MAX-DOAS measurements in an Arctic area, the measured and simulated $O_4$ DSCDs are in good agreement without any correction. Note that the scaling factor mentioned here refers to the ratio between simulated and measured $O_4$ DSCDs, which is opposite to some other studies.

In order to assess whether the $O_4$ DSCD correction is necessary for the MAX-DOAS measurements at the UFS, we compared the measured $O_4$ DSCDs to the simulated ones in the look-up table. Assuming our profile set ($X_{\mathrm{LUT}}$) covers all possible aerosol profiles under cloud-free condition, we derived the $O_4$ scaling factor for each elevation angle and each wavelength based on the statistical analysis. The AODs measured by the sun photometer were used to restrict the range of possible profiles.

Fig. 7 shows the scattered plots of measured and simulated $O_4$ DSCDs of the scanning cycle on 07 Dec 2015 at ∼13:55 UTC. Both the measurements of (a) UV and (b) VIS bands are shown. According to the cloud screening as well as the skycam images, this day was absolutely cloud free. Total AOD measured by the sun photometer at that time is 0.02 and 0.017 for 360 and 477 nm bands, respectively. In each plot, the x-axis indicates the $O_4$ DSCDs measured (or simulated) at the elevation angle of 1°, while the y-axis represents the $O_4$ DSCDs at the other elevation angles. Different colours indicate measurements at different elevation angles. The simulated $O_4$ DSCDs ($\boldsymbol{y}_{\mathrm{s}}(\boldsymbol{x})$) of all the possible profiles in $X_{\mathrm{LUT}}$ are shown as coloured dots. We assume the MAX-DOAS measurement of AOD between 0 and 2 km (denoted as $\tau_{2\mathrm{k}}$, $\tau_{2\mathrm{k}}(\boldsymbol{x}) = 0.5\sigma_1(\boldsymbol{x}) + 0.5\sigma_2(\boldsymbol{x}) + \sigma_3(\boldsymbol{x})$) varies between 50% and 100% of the total AOD measured by the sun photometer (denoted as $\tau_{\mathrm{sp},\lambda}$) in most cases, and the data points of the profiles fulfilling this assumption are highlighted. The measured $O_4$ DSCDs (already corrected for the systematic errors caused by the topography simplification) are plotted as square markers with error bars showing the total uncertainties. It is obvious that at most of the elevation angles, the measured $O_4$ DSCD does not agree with the simulations within the

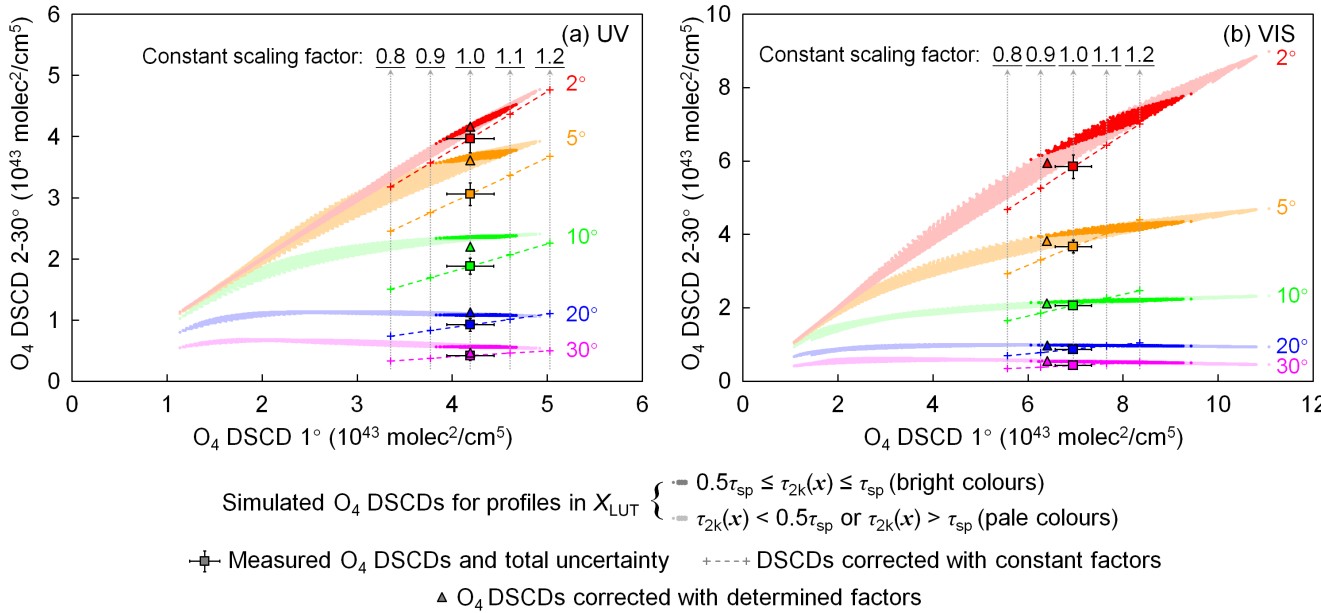

**Figure 7.** Distribution of simulated, measured and corrected $O_4$ DSCDs in (a) UV and (b) VIS bands of the scanning cycle on 07 Dec 2015 at $\sim$13:55 UTC (SZA $\sim$79°, RAA $\sim$39°). X-axes indicate the $O_4$ DSCDs measured (or simulated) at the elevation angle of 1°, while y-axes represent the $O_4$ DSCDs measured (or simulated) at other elevation angles. Different colours indicate measurements at different elevation angles. The coloured dots show the simulated $O_4$ DSCDs of all the possible profiles in the profile set ($X_{\text{LUT}}$). The data points of the profiles with AOD between 0 and 2 km ($\tau_{2k}(\boldsymbol{x})$) varies between 50% and 100% of the total AOD measured by the sun photometer ($\tau_{\text{sp},\lambda}$) are shown in bright colours, while the dots of the other profiles are shown in pale colours. The square markers represent measured $O_4$ DSCDs, and the error bars show the total uncertainties. Systematic errors caused by the topography simplification are already corrected from the measured $O_4$ DSCDs. The plus signs along the dashed lines show the measured $O_4$ DSCDs corrected with constant factors of 0.8, 0.9, 1.1 and 1.2. The triangle markers show the measured $O_4$ DSCDs corrected with the finally determined scaling factors listed in Table 6.

total error. As a result, at both UV and VIS bands, no profiles in $X_{\text{LUT}}$ satisfy the selection requirement ($\chi^2 \leq 9$, see dashed curves in Fig. 8). No profiles matching the measurement is unlikely to happen under such clear sky condition, hence, implies a systematic error and correction of the error is necessary.

5    In order to determine whether the $O_4$ scaling factor is constant for all elevations or dependent on the viewing elevation angles, we first assume it is constant and plot the corrected $O_4$ DSCD measurements in Fig. 7. The plus signs indicate the measured $O_4$ DSCDs corrected with constant scaling factors of 0.8, 0.9, 1.1 and 1.2. Furthermore, the corrected $O_4$ DSCDs should vary along the coloured dashed lines if any other constant scaling factor is applied to the measurements. However, the forward simulation of $O_4$ DSCDs does not overlap with the dashed lines in most of the cases (especially for 5° and 10° of the UV band), indicating that a constant $O_4$ scaling factor for all viewing elevation angles could not resolve the systematic error.

10  Therefore, different scaling factors should be applied to different elevation angles.

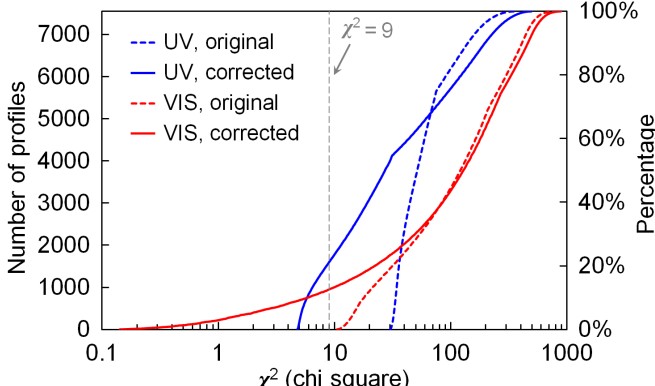

**Figure 8.** Cumulative distribution of the $\chi^2$ of all the profiles in $X_{\mathrm{LUT}}$ for the scanning cycle at 07 Dec 2015 $\sim$13:55 UTC (SZA $\sim$79°, RAA $\sim$39°). Dashed and solid curves refer to the results before and after the $O_4$ DSCD correction, respectively. Blue and red curves refer to the results of the UV and VIS bands, respectively. Note that the x-axis is logarithmically scaled.

In this study, the $O_4$ DSCD scaling factors for each viewing elevation angle and wavelength were determined through the statistical analysis of the long-term observations. We assume the scaling factor mainly depends on the viewing elevation angle, while being less sensitive to other factors such as solar geometry, aerosol load, temperature etc.

Fig. 7 shows that the simulated $O_4$ DSCDs at high elevation angles (e.g. 20° and 30°) vary in a very narrow range. Based on the assumption that $X_{\mathrm{LUT}}$ covers all possible aerosol profiles, the measured $O_4$ DSCDs should lie within the range. The scaling factor can be derived by taking the ratio of the simulated and measured values. As the simulated value varies in a narrow range, the uncertainty of the derived scaling factor should also be low. In order to have better statistics of the scaling factors, this method was applied to the long-term measurements. In addition, only the measurements taken under cloud-free and low aerosol load ($\tau_{\mathrm{sp},\lambda} \leq 0.03$) conditions were used, so as to avoid accounting data contaminated by clouds in the analysis. Here it should be noted that measurements with AOD $< 0.03$ are almost entirely found during winter due to the strong seasonal variation of aerosol load at the UFS. Subsequently, for the wavelength $\lambda$ and the $i^{\mathrm{th}}$ elevation angle of each scanning cycle, we calculate the variation range of the simulated $O_4$ DSCDs for all the profiles fulfilling $0.5\tau_{\mathrm{sp},\lambda} \leq \tau_{2\mathrm{k}}(\boldsymbol{x}) \leq \tau_{\mathrm{sp},\lambda}$, which can be described as a set,

$$Y_{\lambda,i}^* = \{f(\boldsymbol{x}, \lambda, \alpha_i, \theta_i, \phi_i) \mid \boldsymbol{x} \in X_{\mathrm{LUT}}, 0.5\tau_{\mathrm{sp},\lambda} \leq \tau_{2\mathrm{k}}(\boldsymbol{x}) \leq \tau_{\mathrm{sp},\lambda}\}. \tag{10}$$

Only if $\max(Y^*_{\lambda,i}) \leq 1.1 \times \min(Y^*_{\lambda,i})$, then the scanning cycle was taken into account. In most cases, measured $O_4$ DSCDs at high elevation angles are lower than simulated ones, therefore we calculate the scaling factor from the minimum value in $Y^*_{\lambda,i}$ to avoid over-estimation of the scaling factor. The scaling factor derived from this scanning cycle is denoted as

$$\gamma^*_{\lambda,i} = \frac{\min(Y^*_{\lambda,i})}{\Delta S_{\lambda,i}}, \tag{11}$$

where $\Delta S_{\lambda,i}$ is the measured $O_4$ DSCD (already corrected for the systematic errors caused by the topography). For the elevation angles of $5°$, $10°$, $20°$, $30°$ at UV band and $10°$, $20°$, $30°$ at VIS band, numerous scanning cycles from the long-term measurements fulfill the selection criterion, and hence there are sufficient samples of $\gamma^*_{\lambda,i}$ for statistical analysis. We analyzed the frequency distribution of $\gamma^*_{\lambda,i}$ of each elevation and each wavelength band. The result shows that the distributions of $\gamma^*_{\lambda,i}$ follow the normal distribution function with small standard deviation. For instance, for the elevation angle of $20°$, the standard

deviations of UV and VIS bands are both $\sim 0.16$. Subsequently, $\gamma^*_{\lambda,i}$ with the maximum frequency was derived by Gaussian fit. The peak value was used as the scaling factor which is denoted as $\hat{\gamma}_{\lambda,i}$.

    For the low elevation angles ($1°$ and $2°$ at UV band, $1°$, $2°$ and $5°$ at VIS band), as $O_4$ DSCD varies in a wide range, it is impossible to determine the scaling factor with the method mentioned above. However, it is found that in many scanning cycles, within the possible profiles in $X_{\mathrm{LUT}}$, the simulated $O_4$ DSCDs at low elevation angles are well correlated to those at

the neighbouring elevation angle. Therefore, once the scaling factor of the higher elevation angle is determined, we can derive an expected value of the $O_4$ DSCD at the lower elevation angle from the corrected $O_4$ DSCD at the higher one, and the scaling factor can be derived by taking the ratio of the expected value and the measured value.

    For the wavelength $\lambda$ and for each scanning cycle, a subset of $X_{\mathrm{LUT}}$ is defined as

$$X^\dagger = \{ \boldsymbol{x} \mid \boldsymbol{x} \in X_{\mathrm{LUT}}, 0 \leq \tau_{2\mathrm{k}}(\boldsymbol{x}) \leq 2\tau_{\mathrm{sp},\lambda} \}, \tag{12}$$

and the elements of $X^\dagger$ are denoted as $\boldsymbol{x}^\dagger_j$. The corresponding simulated $O_4$ DSCD at the $i^{\mathrm{th}}$ elevation angle is denoted as

$$\Delta S^\dagger_{i,j} = f(\boldsymbol{x}^\dagger_j, \lambda, \alpha_i, \theta_i, \phi_i), \boldsymbol{x}^\dagger_j \in X^\dagger. \tag{13}$$

    A $3^{\mathrm{rd}}$ order polynomial regression is applied between $\Delta S^\dagger_{i,j}$ and $\Delta S^\dagger_{i+1,j}$. The regression function is denoted as $g$. Only if the correlation coefficient $R^2 \geq 0.98$, this scanning cycle is taken into account. As the scaling factor of the $(i+1)^{\mathrm{th}}$ elevation $(\hat{\gamma}_{\lambda,i+1})$ is already determined, the expected value of the $O_4$ DSCD at the $i^{\mathrm{th}}$ elevation angle can be derived with the regression

function:

$$E[\Delta S_{\lambda,i}] = g(\Delta S_{\lambda,i+1} \cdot \hat{\gamma}_{\lambda,i+1}), \tag{14}$$

and the scaling factor derived from this scanning cycle is

$$\gamma_{\lambda,i}^{\dagger} = \frac{E[\Delta S_{\lambda,i}]}{\Delta S_{\lambda,i}}. \tag{15}$$

Similar to the high elevation angles, the frequency distribution of $\gamma_{\lambda,i}^{\dagger}$ from all the available samples was analyzed by fitting to a Gaussian function. The peak value of $\gamma_{\lambda,i}^{\dagger}$ is used as $\hat{\gamma}_{\lambda,i}$. The scaling factor of the $(i-1)^{\text{th}}$ elevation is then derived in the same way. The scaling factors of $1°$ and $2°$ at UV band and $1°$, $2°$, $5°$ at VIS band were determined using this method.

**Table 6.** The finally determined $O_4$ DSCD scaling factors.

| Elevation angle | Factors for corrected $O_4$ DSCDs[a] | | Factors for original $O_4$ DSCDs | |
|:---:|:---:|:---:|:---:|:---:|
| | UV (360 nm) | VIS (477 nm) | UV (360 nm) | VIS (477 nm) |
| 1° | 1.00 | 0.92 | 0.97 | 0.90 |
| 2° | 1.05 | 1.02 | 1.01 | 1.00 |
| 5° | 1.18 | 1.04 | 1.14 | 1.02 |
| 10° | 1.17 | 1.03 | 1.12 | 0.99 |
| 20° | 1.22 | 1.12 | 1.16 | 1.08 |
| 30° | 1.12 | 1.27 | 1.06 | 1.22 |

[a] Means the $O_4$ DSCDs which are already corrected for the systematic errors caused by the topography simplification.

The determined scaling factors are listed in Table 6. The corrected $O_4$ DSCDs are indicated as triangles in Fig. 7. The result shows that except the elevation angle of $1°$, the simulated $O_4$ DSCDs are overestimated comparing to the measured ones. It should be noted that the determination of the scaling factors is based on the measured $O_4$ DSCDs which are already corrected for the systematic errors caused by the topography simplification (discussed in 3.3). Comparing to the original measurements, the result still indicates that the simulated $O_4$ DSCDs at high elevation angles are overestimated. This result is opposite to the results of the other studies. At the moment we have no clear explanation for this finding, it might be related to the specific properties of the high altitude station, e.g. the highly structured topography, horizontal gradients of the aerosol extinction and the systematic dependence of the surface albedo on altitude.

Fig. 8 shows the cumulative distribution of $\chi^2$ of all the profiles in $X_{\text{LUT}}$ for the scanning cycle shown in Fig. 7. The distribution of $\chi^2$ before and after the DSCD correction are shown as dashed and solid curves, respectively. The result indicates that for both UV (blue curves) and VIS (red curves) bands, the $\chi^2$ of most profiles in $X_{\text{LUT}}$ are significantly lower after the correction. As a result, a number of profiles fulfill the selection criterion ($\chi^2 \leq 9$). Note that the AODs measured by MAX-DOAS are still expected to be lower than the sun photometer observations due to the fact that the MAX-DOAS only reports the AOD below 2 km while the sun photometer covers the entire atmosphere.

# 4 Results and discussion

Our retrieval algorithm was applied to the long-term measurement data of the UFS MAX-DOAS from February 2012 to February 2013 and from July 2013 to February 2016. The results are also compared to sun photometer measurements. This section presents the results as well as their discussion.

## 4.1 Dependency of retrieval result on the threshold of cost function

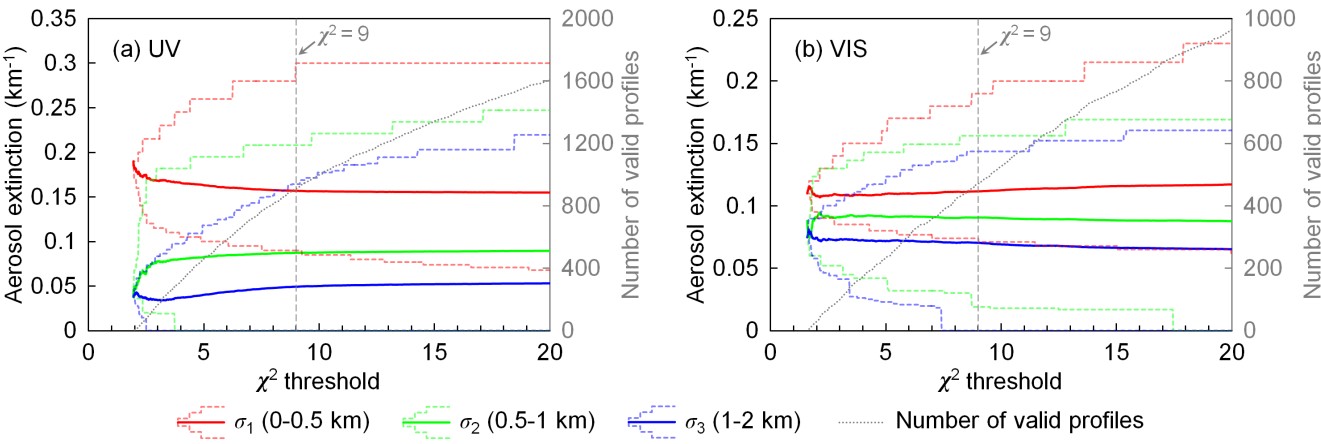

**Figure 9.** Weighted mean profiles, variation ranges of valid profiles and number of valid profiles of (a) UV and (b) VIS bands corresponding to different $\chi^2$ thresholds, results of the scanning cycle on 05 Jul 2015 at $\sim$16:26 UTC (SZA $\sim$64°, RAA $\sim$97°). The weighted mean profiles are shown as solid curves which indicate the aerosol extinction coefficients in the three layers ($\sigma_1$, $\sigma_2$ and $\sigma_3$). The variation ranges of valid profiles are shown as dashed curves which indicates the variation ranges of $\sigma_1$, $\sigma_2$ and $\sigma_3$. The grey dotted curves indicate the number of valid profiles corresponding to different thresholds of $\chi^2$. Measured $O_4$ DSCDs are corrected with the scaling factors listed in Table 6.

As presented in Section 3.8, we consider all the profiles with $\chi^2 \leq 9$ as valid profiles, and the retrieved profile is defined as the weighted mean of all the possible profiles. In this section, we investigate the dependency of the retrieval result on the threshold of $\chi^2$ by comparing the results calculated with different different $\chi^2$ thresholds. Take the two measurement cycles mentioned in Fig. 5 and Fig. 6 for example, Fig. 9 (05 Jul 2015 at $\sim$16:26 UTC) and Fig. 10 (07 Dec 2015 at $\sim$13:55 UTC) show the weighted mean profiles, the variation range of valid profiles and the number of valid profiles corresponding to different $\chi^2$ thresholds. The profiles are shown as coloured curves which indicate the aerosol extinction coefficients in the three layers (i.e. $\sigma_1$, $\sigma_2$ and $\sigma_3$).

The results of both scanning cycles show that the retrieved profile is not sensitive to the threshold of $\chi^2$ when there are sufficient number of valid profiles (number of profiles exceeds $\sim$800 and $\sim$400 for UV and VIS, respectively, see the grey curves in Fig. 9 and Fig. 10). This is because profiles with larger $\chi^2$ have lower weight ($w$). In addition, when the threshold value is increased, more profiles with both higher and lower aerosol extinction coefficients are taken into account. As a result,

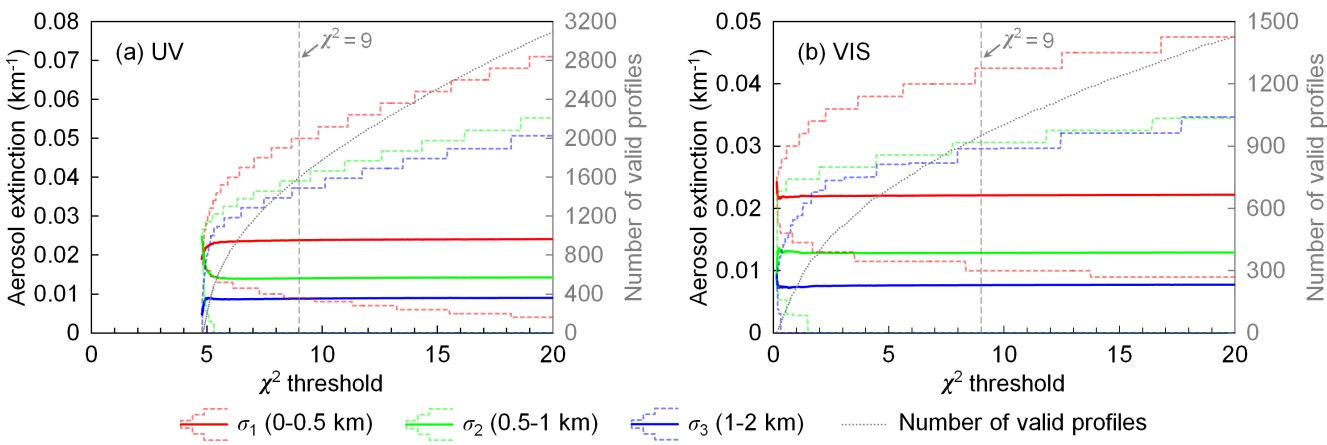

**Figure 10.** Same as Fig. 9, but for the scanning cycle on 07 Dec 2015 at ∼13:55 UTC (SZA ∼79°, RAA ∼39°).

the variation range of valid profiles becomes larger but the weighted mean remains similar. The result shows that the retrieval with a $\chi^2$ threshold of 9 is stable, therefore, it is used in the study.

## 4.2 Estimation of the uncertainties of retrieved profiles

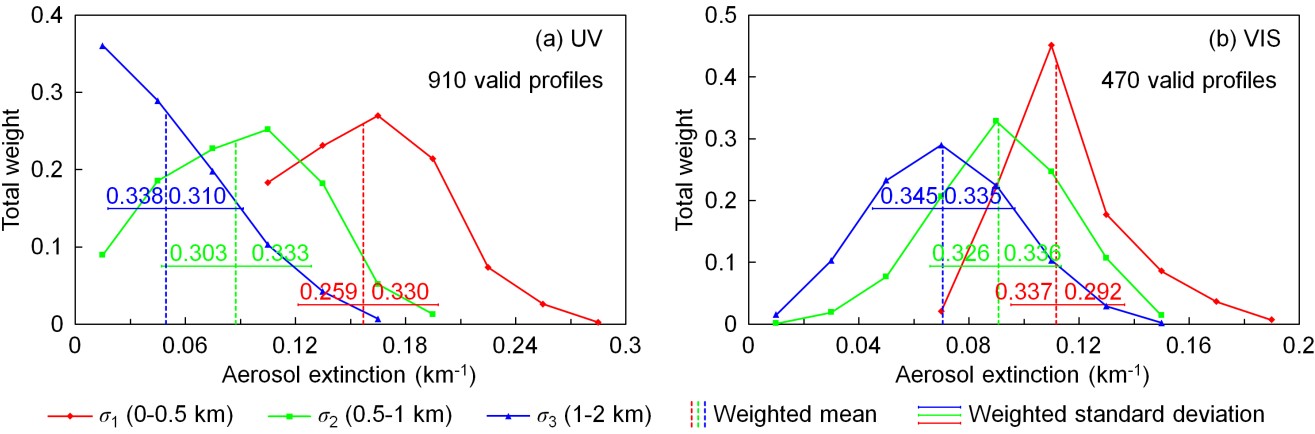

**Figure 11.** Weight distribution of valid profiles of (a) UV and (b) VIS bands, results of the scanning cycle on 05 Jul 2015 at ∼16:26 UTC (SZA ∼64°, RAA ∼97°). The weight distributions of the aerosol extinction coefficients of the three layers ($\sigma_1$, $\sigma_2$ and $\sigma_3$) are shown as solid curves with different colours. The vertical dashed lines indicate the weighted mean aerosol extinction coefficient of the three layers ($\sigma_1(\hat{x})$, $\sigma_2(\hat{x})$ and $\sigma_3(\hat{x})$). The error bars indicate the weighted standard deviation calculated with Eq. (16) and (17). The numbers on the error bars refer to the total weight ($w$) of the profiles covered by each error bar.

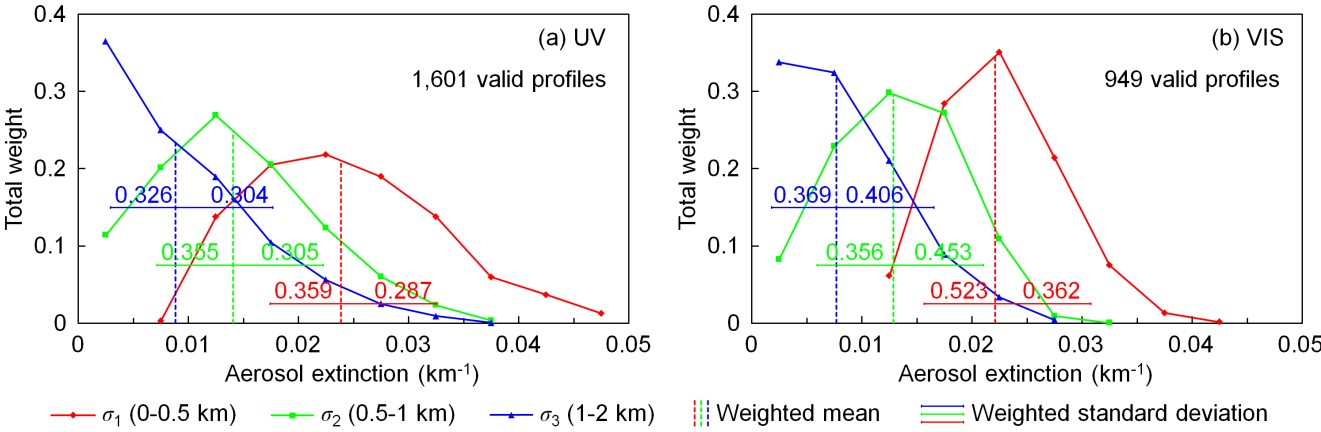

**Figure 12.** Same as Fig. 11, but for the scanning cycle on 07 Dec 2015 at $\sim$13:55 UTC (SZA $\sim$79$^\circ$, RAA $\sim$39$^\circ$).

Still take the two measurement cycles mentioned in Section 4.1 as examples, we analyzed the weight distribution of valid profiles, see Fig. 11 and Fig. 12. The distributions of aerosol extinction coefficients in the three layers ($\sigma_1$, $\sigma_2$ and $\sigma_3$) are shown as solid curves. For each layer, aerosol extinction coefficients of all the valid profiles are grouped, and he y-axis refers to the total weight of each group. The three vertical dashed lines indicate the weighted mean aerosol extinction coefficient of each layer (i.e. $\sigma_1$, $\sigma_2$ and $\sigma_3$ of $\hat{x}$). The result shows that the distributions of $\sigma_1$, $\sigma_2$ and $\sigma_3$ are all asymmetric for both the UV and VIS bands. Especially for the layer of $1-2$ km ($\sigma_3$) at UV band, the weight decreases monotonically with increasing aerosol extinction in both of the two cycles. Take the cycle shown in Fig. 12 (07 Dec 2015 at $\sim$13:55 UTC) as example, there are altogether 205 (12.8%) and 120 (12.6%) valid profiles with $\sigma_3 = 0$ in UV and VIS bands, respectively. These profiles contribute total weights of 0.122 and 0.101 for the UV and VIS retrievals, respectively.

In order to estimate the uncertainty of $\hat{x}$, we calculate the weighted standard deviations of $\sigma_1$, $\sigma_2$ and $\sigma_3$ of all the valid profiles. Due to the asymmetric distribution, the weighted standard deviations are calculated separately for both left (negative) and right (positive) sides. For the $l^{\text{th}}$ ($l = 1, 2, 3$) layer, denote the aerosol extinction coefficient of each profile as $\sigma_l(x)$, then the weighted standard deviation of the left side is calculated from all the valid profiles with $\sigma_l(x) < \sigma_l(\hat{x})$,

$$SD_l^- = \sqrt{\frac{\sum w(x) \cdot [\sigma_l(\hat{x}) - \sigma_l(x)]^2}{\sum w(x)}}, x \in X_{\text{LUT}}, \chi^2(x) \le 1.5M, \sigma_l(x) < \sigma_l(\hat{x}), \tag{16}$$

and the weighted standard deviation of the right side is calculated from all the valid profiles with $\sigma_l(x) > \sigma_l(\hat{x})$,

$$SD_l^+ = \sqrt{\frac{\sum w(x) \cdot [\sigma_l(x) - \sigma_l(\hat{x})]^2}{\sum w(x)}}, x \in X_{\text{LUT}}, \chi^2(x) \le 1.5M, \sigma_l(x) > \sigma_l(\hat{x}), \tag{17}$$

The uncertainties of $\hat{x}$ are indicated as error bars in Fig. 11 and Fig. 12. For each layer, the total weight of the profiles covered by the error bar is labeled in the charts. At the UV band, the total weight of the valid profiles covered by the uncertainties is $59-66\%$, which is close to the standard normal distribution. However, the percentage can be up to 90% at the VIS band. This is because the SNR of the measurement at the VIS band is higher. Therefore the retrieval of VIS band has higher selectivity, and the weight is more concentrated to the mean value.

## 4.3 Retrieval of synthetic measurement data

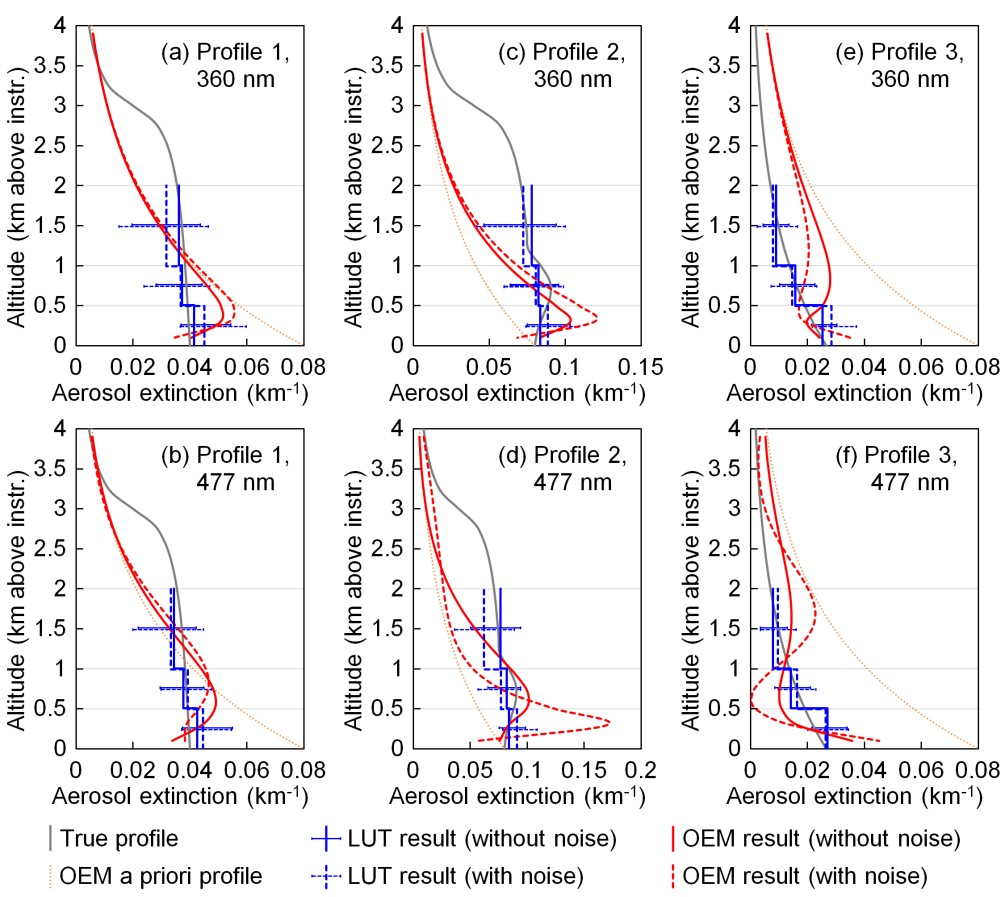

**Figure 13.** Retrieval results of three synthetic profiles. The gray curves show the true profiles, with which the synthetic $O_4$ DSCDs were simulated. The blue and red curves represent the profiles retrieved using our LUT (look-up table) algorithm and a typical OEM (optimal estimation method) algorithm, respectively. The sold blue and red curves represent the profiles retrieved from the original synthetic data, and the dashed curves represent the profiles retrieved from the synthetic data with random noised added. The error bars of the blue curves indicate the uncertainties calculated by Eq. (16) and (17). The dotted orange curves are the a priori profile used in the OEM retrieval.

In order to test the effectiveness of our retrieval algorithm, we generated some synthetic measurement data for the application to our algorithm. Fig. 13 shows the results of three representative synthetic profiles at 360 and 477 nm. In each chart, the true profile is shown as the gray curve. Profile 1 is a tangent curve with aerosols distributed between 0 and 6 km above instrument. The aerosol extinction decreases with increasing altitude, which is $0.04\,\mathrm{km^{-1}}$ at surface level, $\sim$89% at 2 km and 50% at 3 km. The total AOD is 0.12, of which $\sim$92% is contributed from the altitude below 3 km. Profile 2 has a similar shape as Profile 1, but the aerosol extinction between 0.5 and 1 km above instrument was enhanced. The aerosol extinction peaks at 0.75 km, and the average aerosol extinction coefficient between 0.5 and 1 km is larger than the bottom layer by $\sim$10%. In addition, the aerosol extinction coefficients at other altitudes are increased by a factor of 2 comparing to Profile 1. Profile 3 is an exponential profile. The total AOD is 0.12, the scaling height is 1.5 km, and the surface aerosol extinction coefficient is $0.03\,\mathrm{km^{-1}}$.

We first simulated $O_4$ DSCDs at 360 and 477 nm with each profile. The solar position was set as SZA = $60°$ and RAA = $60°$, and the other parameters followed the settings used in calculating the look-up table listed in Table 5 (excluding the aerosol extinction coefficients above 2 km). In order to test the stability of the retrieval, we also generated a set of noisy data for each profile and each wavelength by adding random noise to the simulated $O_4$ DSCDs. We assume the measurement noise at all elevation angles is the same and follows a normal distribution with a standard deviation of 2% of the DSCD of the lowest elevation angle. This noise level is realistic for the measurements at the UFS.

Aerosol profiles were then retrieved from both the original and noisy synthetic data using our algorithm. In the error estimation, the DOAS fitting error ($\epsilon_{\mathrm{fit}}$) was defined as the average values of the UFS measurements, while the other six kinds of errors followed the common settings presented in Section 3.7. $O_4$ DSCD correction was not applied. The solid and dashed blue curves in Fig. 13 show the profiles retrieved from the original and noisy data, respectively, and the error bars indicate the uncertainties calculated by Eq. (16) and (17). The results show that for Profile 1 and Profile 3, our retrieval algorithm can well reproduce the true profiles from not only the original data but also the noisy data. For Profile 2, the retrieved profile cannot reproduce the elevated layer, but the error bar covers the aerosol extinction of the true profile. This is because the retrieval is ill-posed, which means the limited input information does not correspond to a unique profile with elevated layer, instead, many other profiles without the elevated layer can also fit the input information. Adding noise to the synthetic data can affect the retrieved aerosol extinction coefficients, however the influence is small in most cases. In addition, the noise can amplify the uncertainty of retrieved profile. The results indicate that our LUT-based retrieval is stable.

We also retrieved the synthetic data using the bePRO profiling tool developed by BIRA-IASB (Clémer et al., 2010; Hendrick et al., 2014). It is an OEM-based algorithm and uses LIDORT as the forward model. In the retrieval of all the six cases, the a priori profile was defined as an exponential profile with AOD = 0.12 and scaling height = 1.5 km, shown as the dotted orange curve in each panel of Fig. 13. The vertical grid was defined as 20 layers of 200 m thickness each. For Profile 1 and Profile 2, the uncertainty covariance matrix of a priori ($\mathbf{S}_a$) was defined as in Clémer et al. (2010) and Wang et al. (2014a): the diagonal elements corresponding to the bottom layer, $\mathbf{S}_a(1, 1)$, was set as the square of a scaling factor $\beta$ ($\beta$ = 0.2) times the maximum partial AOD of the profiles; the other diagonal elements decrease linearly with increasing altitude to $0.2 \times \mathbf{S}_a(1, 1)$; the off-diagonal elements of $\mathbf{S}_a$ were defined using Gaussian functions with correlation length $\gamma$ = 0.05 km. For Profile 3, as the difference between the true and a priori profiles is quite large, we set $\beta$ = 0.4 and $\gamma$ = 0.1 km, so that the constrain from

the a priori profile is weaker. The measurement uncertainty covariance matrix ($\mathbf{S}_\epsilon$) was also defined as in most of the other MAX-DOAS studies that $\mathbf{S}_\epsilon$ is a diagonal matrix with variances equal to the square of the DOAS fitting error ($\epsilon_{\text{fit}}^2$). We defined $\epsilon_{\text{fit}}$ as same as in the LUT retrieval, but the other six error sources were not included. The retrieval parameters related to the radiative transfer simulation followed the settings of our LUT-based retrieval.

5    The results retrieved from the data with and without noise are shown in Fig. 13 as solid and dashed red curves, respectively. In all the 12 retrieval cases, the O$_4$ DSCDs simulated with retrieved profiles are well correlated to the input values (the relative root mean square error varies between 0.7% and 4.7%). However, as the retrieval is ill-posed, the retrieved profiles cannot well reproduce the true profile. Especially at high altitudes (above 1 km), the retrieved profiles are mostly dominated by the a priori profile. The OEM retrieval is also sensitive to measurement noise, which can be seen from the large variations of profile shape 10   and aerosol extinction. The results indicate that the LUT based algorithm is much more suitable for measurements with low SNR.

## 4.4 Comparison to sun photometer measurements

Fig. 14 shows the comparison of AOD measured by MAX-DOAS and sun photometer during the entire study period. The seasonally averaged AOD measured by both instruments are listed in Table 7. As the AOD measured by MAX-DOAS refers to 15   the AOD between 0 and 2 km while the AOD measured by sun photometer refers to the total AOD, the sun photometer results should be larger. Despite of the difference, the time series (panels (a) and (c) of Fig. 14) show that the AODs measured by both instruments have a similar seasonal variation with higher AOD in summer and lower AOD in winter. The monthly average data show that the difference between the AODs measured by MAX-DOAS and sun photometer is much larger in summer, this coincides with the ceilometer profiles shown in Fig. 1 which indicate much higher aerosol extinction coefficients above 2 km 20   in summer. The underestimation of MAX-DOAS may also be related to the decreased sensitivity of measurement at higher altitudes.

**Table 7.** Seasonally averaged AODs measured by the MAX-DOAS and sun photometer at the UFS. The AODs measured by MAX-DOAS refer to the AODs between 0 and 2 km above instrument (i.e. 2,650 − 4,650 m a.s.l.), and the measurements were available during the daytime with SZA < 85° and no cloud; the AODs measured by sun photometer refer to the total AOD, and the measurements were only available during 10:00 − 14:00 UTC. The results listed in the table are calculated from all the available hourly averaged AODs.

| Season | AOD (0 − 2 km) measured by MAX-DOAS | | Total AOD measured by sun photometer | |
|---|---|---|---|---|
| | 360 nm | 477 nm | 360 nm | 477 nm |
| Spring (MAM) | 0.064 | 0.065 | 0.106 | 0.101 |
| Summer (JJA) | 0.121 | 0.114 | 0.214 | 0.184 |
| Autumn (SON) | 0.048 | 0.040 | 0.070 | 0.068 |
| Winter (DJF) | 0.028 | 0.024 | 0.037 | 0.033 |

The correlation between hourly averaged AODs measured by MAX-DOAS and sun photometer is shown in Fig. 14 (b, d). AODs show a general agreement at the UV and the VIS bands with correlation coefficients of $R = 0.733$ and 0.798, respectively.

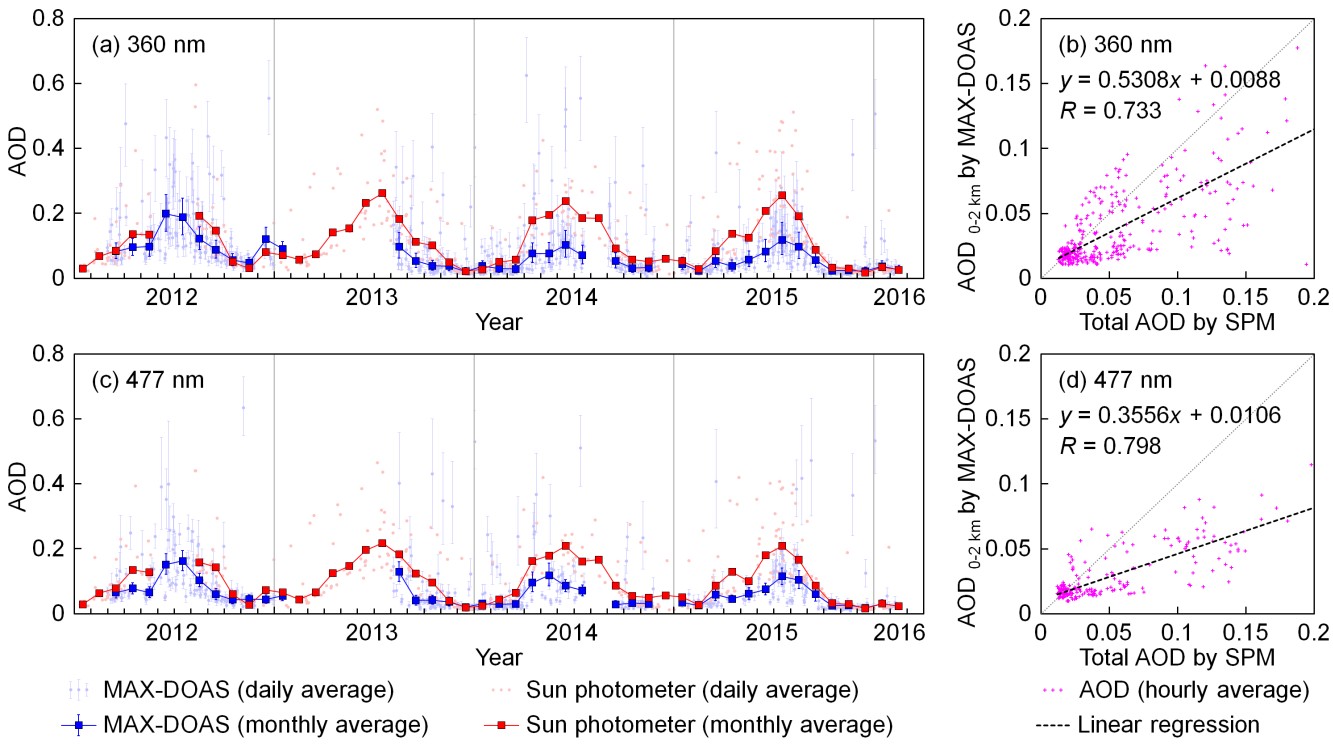

**Figure 14.** Comparison of AODs at (a, b) 360 nm and (c, d) 477 nm measured by MAX-DOAS and sun photometer. The charts on the left side (a, c) show the daily and monthly averaged time series, whereas the scatter plots on the right side (b, d) show the hourly averaged results. The AOD measured by MAX-DOAS refers to the vertical range between 0 and 2 km (i.e. $\tau_{2k}(\hat{x}) = 0.5\sigma_1(\hat{x}) + 0.5\sigma_2(\hat{x}) + \sigma_3(\hat{x})$) above the instrument (i.e. $2,650 - 4,650$ m a.s.l.). The measurements were available during daytime with SZA $< 85°$ and cloud-free conditions. The AOD measured by the sun photometer refer to the total AOD, and only the measurements during $10:00 - 14:00$ UTC were used for reasons of their accuracy. The daily and monthly averaged results are calculated from all the available hourly averaged AODs, therefore they are not real monthly and daily averages. The error bars of the MAX-DOAS data refer to the averages of the uncertainties calculated by Eq. (16) and (17). A few data points are outside the scatter plots.

However, AODs from MAX-DOAS are lower, consequently the slope of the regression lines are 0.5308 and 0.3556 for UV and VIS bands, respectively. As the MAX-DOAS only reports AODs below 2 km while the sun photometer measures the total AODs, the MAX-DOAS AODs are indeed expected to be lower. This is in particular true in cases of large AODs due to very strong convection of polluted air masses from the valley and/or the presence of Saharan dust layers. Then, particles are often transported beyond the range of the MAX-DOAS measurements and the disagreement is largest. This feature might be strengthened by the decreased sensitivity of the MAX-DOAS measurement at higher altitudes, so that the upper part of an aerosol layer is missed. In addition, a few data points lie above the 1:1 reference lines. This might be explained by the inhomogeneous distribution of aerosols in horizontal direction. The light paths of the MAX-DOAS and the sun photometer

are different. MAX-DOAS measures scattered sunlight while sun photometer derives the AOD from direct sun measurements. Therefore, when the aerosol load along the light path of MAX-DOAS is higher than that of the direct sun measurement, the AOD measured by the MAX-DOAS may exceed the AOD measured by the sun photometer. For most of these points, the difference between the results of the two instruments is within their uncertainty ranges, i.e., the disagreement is probably due to the measurement and retrieval errors.

## 4.5   Temporal variation of aerosol characteristics

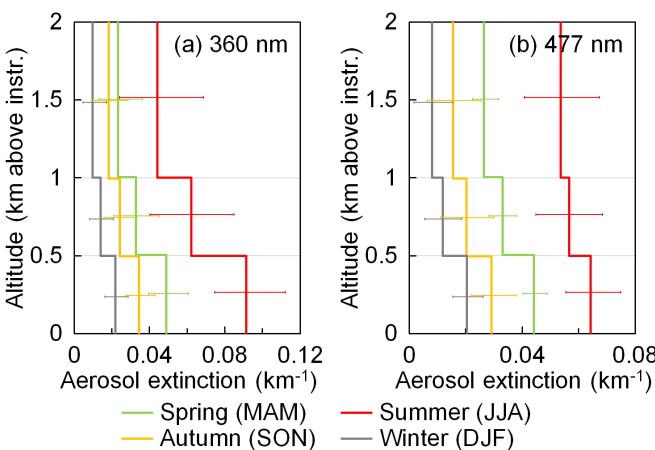

**Figure 15.** Seasonal average aerosol extinction profiles for (a) 360 and (b) 477 nm derived from the long-term measurement results. The error bars represent the averages of the uncertainties calculated by Eq. (16) and (17).

The seasonally averaged aerosol extinction profiles derived from the long-term measurements are shown in Fig. 15. The result indicates that the aerosol load is high in summer and low in winter, which coincides with the ceilometer results shown in Fig. 1. The seasonal pattern can be explained by the higher biogenic emissions from vegetation in summer. Moreover, the mixing layer is higher in summer, thus anthropogenic aerosols are more likely dispersed to upper altitudes. The shape of the profiles also agree with the ceilometer results that the averaged aerosol extinction decreases with increasing altitude in all seasons – taking into account the coarse vertical resolution of the MAX-DOAS. In addition, Fig. 15 shows a much larger vertical gradient at 360 nm in summer. This might be explained by the lower sensitivity of the UV measurement for higher altitudes due to the more decreased visibility at shorter wavelength.

We compared the seasonally averaged aerosol extinction coefficients at 360 and 477 nm in the bottom layer ($0 - 0.5$ km above the instrument, $\sigma_1$), see Fig. 16. The averaged aerosol extinction coefficients are shown as bar charts. The ratio between the aerosol extinction coefficients at 360 and 477 nm is indicated by the grey curve. The result shows that the aerosol extinction coefficient ratio between 360 and 477 nm is significantly higher in summer than in the other seasons.

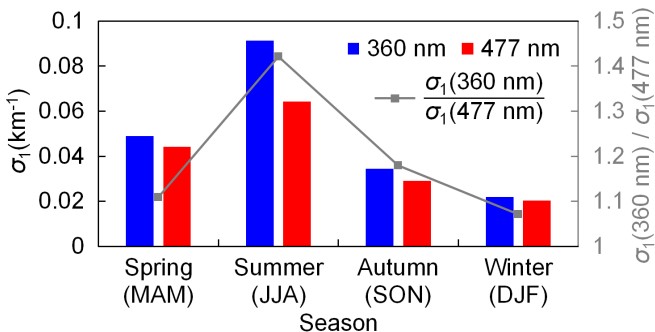

**Figure 16.** Comparison of seasonal average aerosol extinction coefficients at 360 and 477 nm in the bottom layer ($0-0.5$ km above instrument, $\sigma_1$). The coloured bars show the average aerosol extinction coefficients of the four seasons (equal to the bottom values shown in Fig. 15). The grey square markers indicate the ratios between the aerosol extinction coefficients at 360 and 477 nm.

From these ratios the Ångström exponent (AE) can be calculated using the seasonally averaged surface aerosol extinction coefficients at 360 and 477 nm. The results are listed in Table 8. The seasonal averaged AEs of $380-500$ nm from the AERONET measurements at Hohenpeißenberg from April 2013 to February 2016 are also listed for comparison. The result shows that both the UFS and Hohenpeißenberg measured the highest AE in summer and the lowest in winter. The AE at the UFS is in general lower than that measured at Hohenpeißenberg with a smaller difference in summer. This can be explained by the different altitude of the two sites. As the AERONET station at Hohenpeißenberg is located at $\sim$950 m a.s.l., larger contribution of anthropogenic aerosols is expected. The extremely low AE at the UFS in spring, autumn and winter agrees with the result measured at a plateau site (Lhasa, China, 3,688 m a.s.l.) reported in Xin et al. (2007). The annual mean AE at that site is reported to be $0.06 \pm 0.31$, which is significantly lower than those measured at low-altitude sites, especially urban and forest sites. In general, a smaller AE implies larger aerosol particle sizes (Dubovik et al., 2002). The increased AE at UFS in summer indicates a larger contribution of fine particles. The result is consistent with the fact that the particle size of biogenic secondary aerosols is in general smaller than ice particles transported from the lower altitudes to upper altitudes in summer.

**Table 8.** Seasonal average Ångström exponents (AEs) obtained from MAX-DOAS near-surface measurements ($0-0.5$ km above instrument) and from AERONET measurements at Hohenpeißenberg. The results of MAX-DOAS are calculated from the ratios between the seasonal average aerosol extinction coefficients at 360 and 477 nm (i.e. the ratios shown in Fig. 16). The results of AERONET are the seasonal average values of AEs ($380-500$ nm) at Hohenpeißenberg from Apr 2013 to Feb 2016.

| Season | AE from UFS MAX-DOAS | AE from AERONET at Hohenpeißenberg |
|---|---|---|
| Spring (MAM) | 0.37 | 1.26 |
| Summer (JJA) | 1.25 | 1.38 |
| Autumn (SON) | 0.59 | 1.05 |
| Winter (DJF) | 0.24 | 0.47 |

## 5 Summary and conclusions

We have developed a new MAX-DOAS aerosol profile retrieval algorithm based on a parameterized $O_4$ DSCD look-up table. The algorithm is applied to the long-term MAX-DOAS measurements at the UFS, Germany, a high altitude site located at 2,650 m a.s.l.

Observations of $O_4$ absorptions at both 360 and 477 nm were analyzed. We first investigated the sensitivities of $O_4$ absorption to several parameters. According to the sensitivity analysis result, we defined an aerosol profile set which consists of 7,553 possible profiles and then simulated $O_4$ DSCDs with all the profiles and all possible observation geometries. In the retrieval of each measurement cycle, the simulated $O_4$ DSCDs corresponding to all the possible profiles are obtained from the look-up table. The cost functions ($\chi^2$) are calculated for each possible profile according to the simulated and measured $O_4$ DSCDs as well as the measurement uncertainties. A comprehensive error analysis is performed to estimate the total uncertainty. Valid profiles are selected from the profile set according to the cost function. The optimal solution is defined as the weighted mean of the valid profiles.

One key result of our study is that an elevation dependent $O_4$ DSCD scaling factor is needed to bring measured and simulated $O_4$ DSCDs into agreement. Assuming the look-up table covers all possible aerosol profiles under clear sky conditions, we determined the scaling factors based on the statistical analysis of the long-term measurements. The agreement between measured and simulated $O_4$ DSCDs is greatly improved by this correction.

In addition, we developed a simple cloud screening method which is based on the statistical analysis of the colour index. The developed cloud screening method is applied to the long-term measurements to filter out data taken under cloudy conditions.

In order to test the effectiveness of the algorithm, we retrieved profiles from synthetic data. The results indicate that our algorithm can well reproduce the true profile, and the retrieval is stable to measurement noise.

The AOD retrieved from the long-term MAX-DOAS measurements was compared to the sun photometer observations at the UFS. The results show reasonable agreement with each other. However, especially in summer the sun photometer results are systematically larger (by about a factor of 2) than the MAX-DOAS results. This discrepancy is due to the different definitions of AOD measured by MAX-DOAS and sun photometer. The larger difference in summer also coincides with the ceilometer measurements at the UFS which indicate larger aerosol extinctions at high altitude in summer. The long-term observation results show that the aerosol load at the UFS is higher in summer and lower in winter. Higher AOD in summer is mainly related to a higher frequency of extended mixing layers that allows particles to disperse from lower to upper altitudes. According to the MAX-DOAS measurements, the mean aerosol extinction decreases with increasing altitude for all seasons, this agrees with the ceilometer measurements. The Ångström exponent derived from MAX-DOAS surface measurement is higher in summer and extremely low in winter, which implies a smaller particle size in summer. This might be due to a significant contribution from biogenic sources in summer.

The study demonstrated that the developed method is effective for MAX-DOAS measurements at the UFS. Since the profile set only consists of reasonable profiles and the final solution is calculated from the weighted mean of all valid profiles, and from the fact that the retrieval does not rely on a priori profiles, many of the limitations of retrieval algorithms based on the

optimal estimation method are overcome. In addition, as the $O_4$ DSCDs of all possible profiles are pre-calculated, our method significantly reduces the computational time, so that real-time retrievals should be possible.

## Appendix A:  Cloud screening method

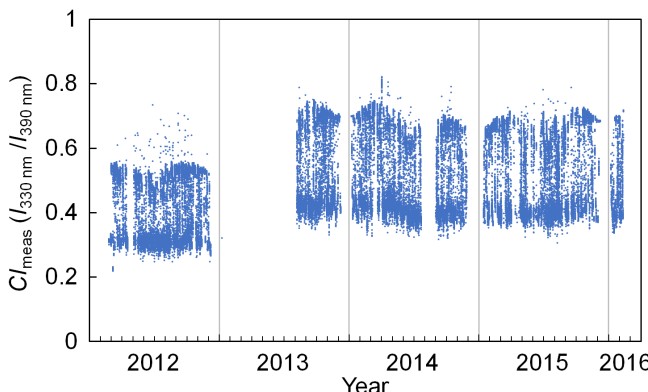

**Figure A1.** Time series of $CI_{\mathrm{meas}}$ calculated from the zenith UV spectra measured during the entire study with $30° < \mathrm{SZA} < 70°$.

In this study, the colour index (CI) is defined as the ratio of radiative intensities at 330 and 390 nm. Measured CIs (denoted

as $CI_{\mathrm{meas}}$) were calculated from the zenith UV spectra (offset and dark current corrected) by taking the ratio of the counts at 330 and 390 nm. Fig. A1 shows the time sereies of $CI_{\mathrm{meas}}$ calculated from all the zenith spectra with $30° < \mathrm{SZA}$ (solar zenith angle) $< 70°$ during the entire study. The result shows that the variation range of $CI_{\mathrm{meas}}$ is stable within the two periods. However, the optical throughput of the instrument in the UV spectral range has been enhanced after a regular maintenance of the optical system in 2013. Hence, the CI increased systematically in the second period. Therefore, calibration of $CI_{\mathrm{meas}}$ is

necessary in order to make the $CI_{\mathrm{meas}}$ measured during the two periods comparable to each other. The calibration was done following the method suggested in Wagner et al. (2016). $CI_{\mathrm{meas}}$ measured under overcast skies were fitted to the simulated minimum CI. The correction factor was determined to be 2.70 and 2.06 for the periods of $02.2012 - 01.2013$ and $08.2013 - 02.2016$, respectively. $CI_{\mathrm{meas}}$ was subsequently converted to $CI_{\mathrm{cal}}$ (calibrated CI) by multiplying the corresponding correction factor.

Fig. A2 shows the frequency distribution of $CI_{\mathrm{cal}}$ measured with different SZAs. The $CI_{\mathrm{cal}}$ from the long-term measurements were grouped by their SZA with a step size of $2°$. The relative frequency distributions are colour coded. The result shows a bimodal frequency distribution of $CI_{\mathrm{cal}}$ for all SZAs. The peaks with lower and higher CI are corresponding to measurements under overcast and clear skies, respectively. This pattern is similar to the CI measured on Jungfraujoch, Switzerland (3570 m a.s.l.) reported in Gielen et al. (2014), and different from the results measured at the low-altitude sites reported in Gielen et al.

(2014); Wagner et al. (2016). This is because the high altitude sites are seldom influenced by anthropogenic aerosols, hence the sky is either clear or covered by cloud or fog during most of the time. Based on this feature, we defined the threshold for

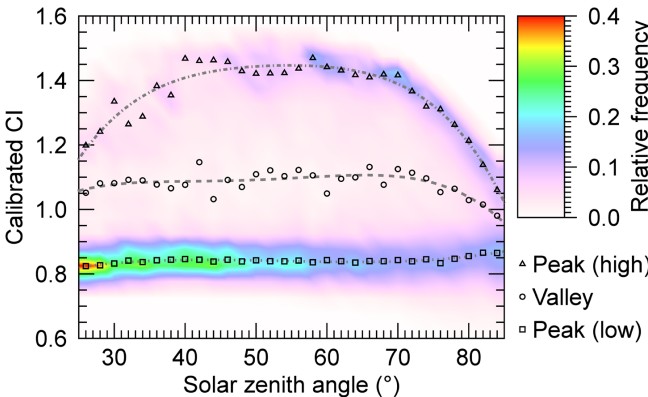

**Figure A2.** Distribution pattern of $CI_{cal}$ during the entire study. Data were grouped by SZA with an interval of $2°$. For each group, frequency was counted for bins of 0.05. Peak and valley values (shown as markers) were determined by Gaussian fit. The curves are the results of $4^{th}$ order polynomial regressions of each data series.

cloud screening as the $CI_{cal}$ with the minimum probability between the two peaks (denoted as $CI_{cal, valley}$). The $CI_{cal, valley}$ was determined by fitting the probability density function to a Gaussian function. The circle markers shown in Fig. A2 indicates the determined $CI_{cal, valley}$. In order to minimize the noise, the $CI_{cal, valley}$ was fitted to a $4^{th}$ order polynomial. The resulting smoothed $CI_{cal, valley}$ was used as the threshold (indicated as dashed curve in Fig. A2). Based on this approach, $\sim$60% of the zenith measurements were determined as cloudy scenes, and the corresponding scanning cycles were not used in the following analysis.

## Appendix B: Result of the sensitivity studies

We investigated the sensitivity of $O_4$ absorption to surface albedo, single scattering albedo (SSA), scattering phase function, aerosol extinction at different altitude, aerosol extinction above retrieval height, and surface aerosol extinction. In the test for each parameter, $O_4$ DSCDs at 360 and 477 nm and at the six off-zenith elevations were simulated with the parameter being tested set as different values, while all the other parameters were fixed. In this section, we only present the results of the sensitivity tests under the common settings listed in Table B1. In the following subsections, all the unmentioned simulation parameters followed the common settings. The extreme and median values of each parameter are also discussed in the following subsections.

### B1 Sensitivity to surface albedo

It is difficult to estimate the surface albedo around the measurement site. In other studies, the surface albedo at low altitude sites was usually estimated to be $0.05 - 0.1$ (e.g., Irie et al., 2008; Ma et al., 2013; Wagner et al., 2011; Chan et al., 2017; Li et al., 2010; Clémer et al., 2010; Li et al., 2013; Wang et al., 2016), while at a high altitude site, it was estimated to be 0.2 (Franco

**Table B1.** The common settings for sensitivity studies.

| Parameter | Value or definition |
|---|---|
| Topography | Flat surface at an altitude of 2,650 m a.s.l. |
| Solar zenith angle (SZA) | 60° |
| Relative solar azimuth angle (RAA) | 60° |
| Surface albedo | 0.1 |
| Single scattering albedo (SSA) | 0.93 (360 nm) / 0.92 (477 nm) |
| Phase function | The 'median' phase function defined in Appendix B3 |
| Aerosol extinction profile | Box-shape profile with AOD = 0.12 and box-height = 3 km (i.e. $\sigma = 0.04\,\mathrm{km}^{-1}$ for 2.65 – 5.65 km a.s.l. and $\sigma = 0$ for altitude > 5.65 km) |
| Climatology | US standard profiles for profile, temperature and trace gas profiles |

et al., 2015). As for the UFS, on one hand, the snow covers and naked rocks are more reflective than the typical urban and rural surfaces; on the other hand, the deep valleys close to the site can significantly decrease the surface albedo. In addition, the measurements at different elevation angles are sensitive to different parts of surface. The effective surface albedo also depends on the observation geometry. The forming and melting of the snow cover can affect the surface albedo as well. However, the RTM can only assume a constant surface albedo. Therefore, we have to estimate a variation range of the surface albedo and consider the possible uncertainty in the retrieval. In this study, we empirically estimate that the surface albedo varies between 0.025 and 0.2 with a median value of 0.1 for both 360 and 477 nm.

In order to estimate the uncertainty of simulated $O_4$ DSCD due to the surface albedo, we simulated $O_4$ DSCDs with extreme surface albedo values (0.025 and 0.2) and the median value (0.1), while the other parameters were fixed as the settings listed in Table B1. Besides the box-shape profile with AOD = 0.12, we also did a test with an aerosol-free profile. The relative differences of the $O_4$ DSCDs simulated with extreme surface albedo values compared to those simulated with the median value are shwon in Fig. B1.

The result shows that at both 360 and 477 nm, $O_4$ DSCDs of all elevation angles slightly decrease with increasing surface albedo, and the variation rate differs with different elevation angles and different aerosol loads. Based on our estimation of the variation range of surface albedo, if the estimated median value (0.1) is used in the forward simulation, the uncertainty caused by the surface albedo assumption would be less than 3%, and the positive and negative errors are nearly equal. Our further simulations also show that the uncertainty caused by surface albedo depends on the observation geometry. In the aerosol profile retrieval, we use a simple look-up table to determine the simulation error caused by surface albedo (see Section 3.7.2).

**B2  Sensitivity to single scattering albedo**

As aerosol optical property data at the UFS are not available, and we use the AERONET data at Hohenpeißenberg instead. According to the long-term data, for the single scattering albedo (SSA) at 360 nm, 90% of the data vary between 0.87 and

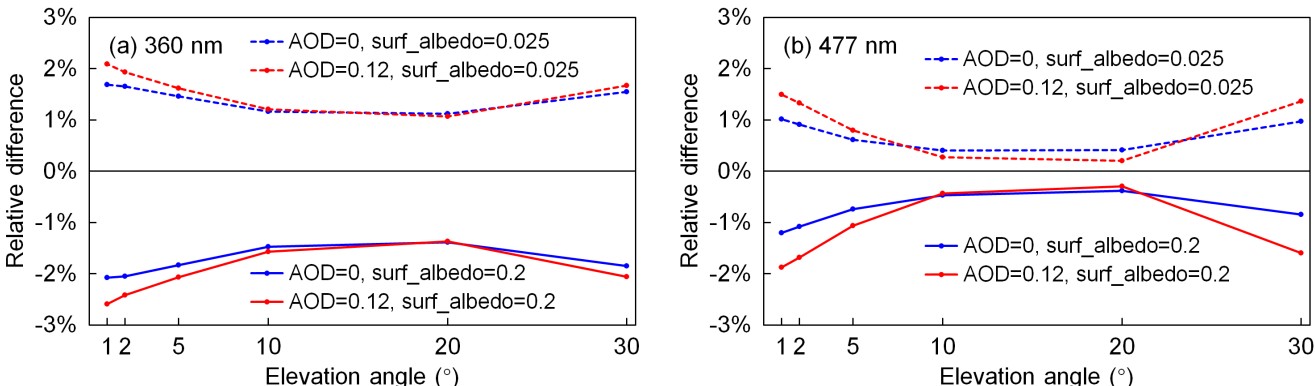

**Figure B1.** Relative differences of $O_4$ DSCDs at (a) 360 nm and (b) 477 nm simulated with extreme surface albedo values (solid lines for 0.2 and dashed lines for 0.025) compared to $O_4$ DSCDs simulated with the median value (0.1). Blue lines refer to the results under aerosol-free condition, while red lines refer to the results with a box-shape profile with AOD = 0.12 and box height = 3 km. The other simulation parameters followed the settings listed in Table B1.

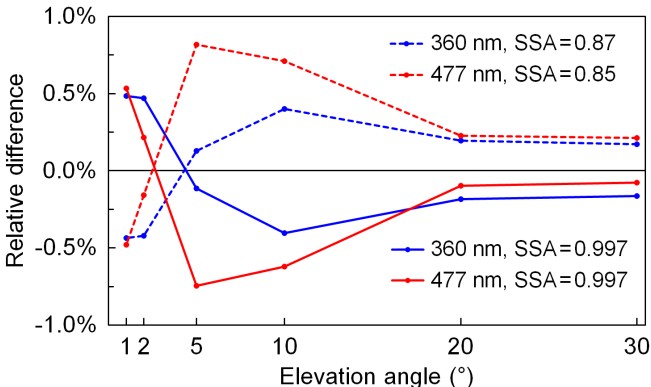

**Figure B2.** Relative differences of $O_4$ DSCDs at 360 nm (blue lines) and 477 nm (red lines) simulated with extreme single scattering albedo values (solid lines for larger extreme value and dashed lines for smaller extreme value) compared to $O_4$ DSCDs simulated with the median value (0.93 for 360 nm and 0.92 for 477 nm). The other simulation parameters followed the settings listed in Table B1.

0.997, and the median value is 0.93; for the SSA at 477 nm, 90% of the data vary between 0.85 and 0.997, and the median value is 0.92.

In order to estimate the uncertainty of simulated $O_4$ DSCD due to the SSA, $O_4$ DSCDs were simulated with the median and extreme SSA values (0.87, 0.93 and 0.997 for 360 nm; 0.85, 09.2 and 0.997 for 477 nm), while the other parameters were fixed as the settings listed in Table B1. The relative differences between the $O_4$ DSCDs simulated with extreme and median SSA values are shown in Fig. B2.

The result indicates that using the median SSA in the forward simulation would result in less than 1% error in $O_4$ DSCDs in 90% of the cases. In addition, the positive and negative errors are mostly equal. Although the measurements of SSA were taken at a much lower altitude site, the sensitivity result shows the error attributed to SSA is rather small (<1%). Therefore, using the SSA values from Hohenpeißenberg should not have a big influence on the retrieval. Since the simulation error caused by SSA can be influenced by the aerosol load as well as the observation geometry, it is determined using a simple look-up table in our aerosol profile retrieval (see Section 3.7.2).

## B3   Sensitivity to scattering phase function

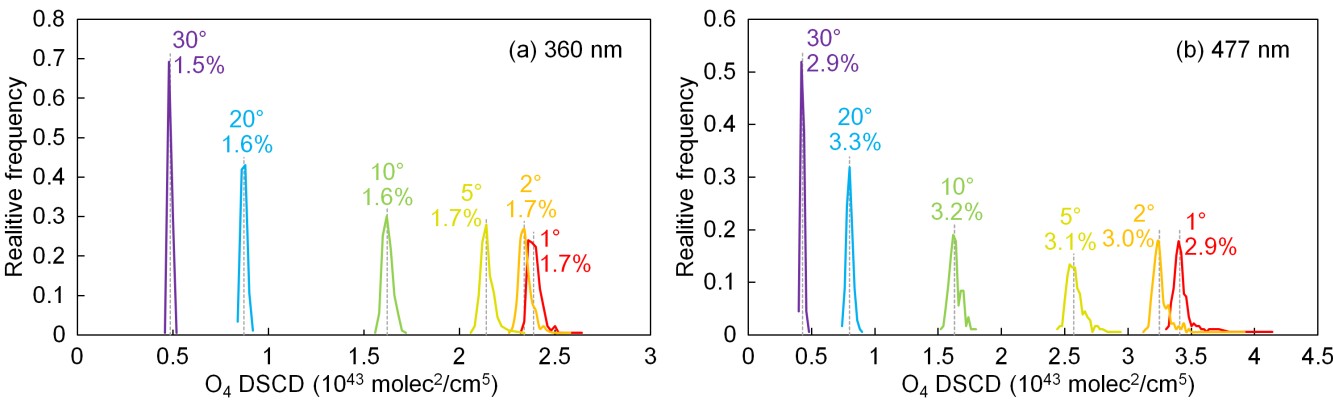

**Figure B3.** Frequency distribution of $O_4$ DSCDs at (a) 360 nm and (b) 477 nm simulated with all the phase functions during 2013 – 2014. The other simulation parameters followed the settings listed in Table B1. The percentage standard deviations of the simulated $O_4$ DSCDs at each elevation angle are labeled in the plots. The grey dashed lines represent the median values of simulated $O_4$ DSCDs at each elevation angle.

The estimation of the uncertainty of simulated $O_4$ DSCD due to scattering phase function is also based on the AERONET data at Hohenpeißenberg. Unlike most of the other simulation parameters which can be defined by a single number, the parameter of scattering phase function is defined by function values at different scattering angles. In order to estimate the uncertainty, we simulated $O_4$ DSCDs with all the phase functions from 2013 to 2014 (altogether 179 available data), while the other parameters were fixed as the settings listed in Table B1. The frequency distributions of simulated $O_4$ DSCDs are shown in Fig. B3. For each elevation angle, the percentage standard deviation is indicated beside the curve, and the gray dashed line shows the median value. The results indicate that the distribution of the simulated $O_4$ DSCDs follow the normal distribution, and the standard deviation at 477 nm is larger comparing to 360 nm.

Based on the simulation results, the phase function with which the simulated $O_4$ DSCDs at all elevation angles are closest to the median values is chosen as the so-called 'median' phase function for each wavelength. In our aerosol profile retrieval, the error caused by scattering phase function is also determined using a simple look-up table (see Section 3.7.2).

## B4 Sensitivity to aerosols at different altitude

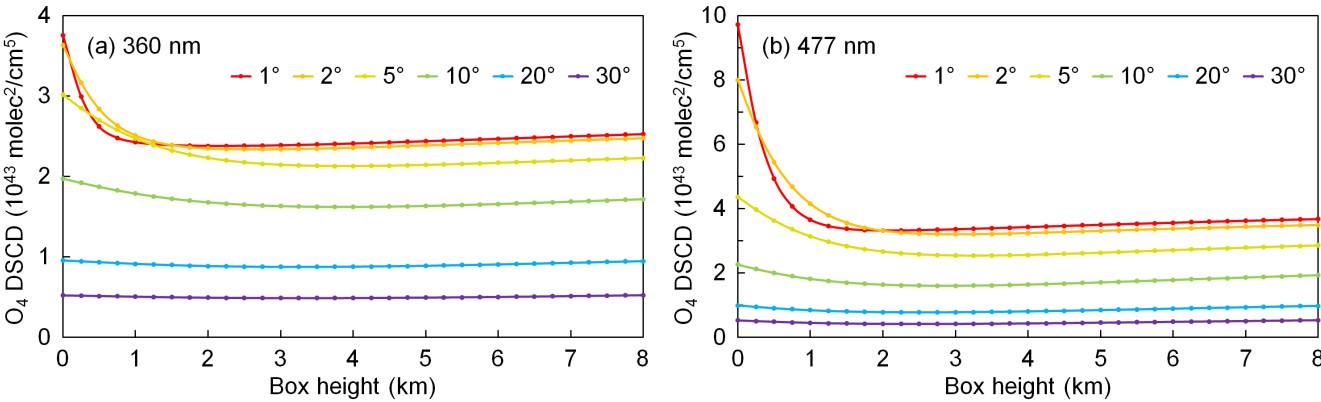

**Figure B4.** Simulated $O_4$ DSCDs at (a) 360 nm and (b) 477 nm for box-shape profiles with the same surface aerosol extinction coefficient of 0.04 km$^{-1}$ and different box heights from 0 to 8 km. The other simulation parameters followed the settings listed in Table B1.

The sensitivity of $O_4$ DSCD to aerosol extinction at different altitude was estimated by simulating $O_4$ DSCDs with box-shape aerosol profiles with the same aerosol extinction coefficient of 0.04 km$^{-1}$ and different box heights varying from 0 to 8 km. The other parameters were fixed as the settings listed in Table B1. Fig. B4 shows the simulated $O_4$ DSCDs at 360 and

477 nm for each elevation angle. The result indicates that the sensitivities of $O_4$ DSCDs at all elevation angles decrease rapidly with increasing box height (and also increasing AOD). Furthermore, $O_4$ DSCDs at all elevation angles are almost constant when the box height varies between 2 and 8 km, which indicates that $O_4$ absorption is almost insensitive to the aerosols above 2 km. Take the $O_4$ DSCD measured at 360 nm with elevation angle of 2° as an example, the sensitivity to aerosols at 2 km is lower than that at surface level by a factor of $\sim$40. In addition, measurements at lower elevation angles are more sensitive to

aerosols close to the surface compared to higher elevations. According to the result, our retrieval of aerosol profiles would only focus on aerosols below 2 km.

This result coincides with the results reported in the MAX-DOAS studies based on the OEM (e.g., Frieß et al., 2006; Clémer et al., 2010; Frieß et al., 2016; Bösch et al., 2018). In these studies, the averaging kernels — which indicate the measurement sensitivity to aerosols at different altitude — are all close to zero at the altitudes above 2 km.

**B5 Sensitivity to aerosols above retrieval height**

As discussed in Section B4, our aerosol profile retrieval would only focus on aerosols below 2 km. However, as the aerosol load on Zugspitze is usually very low and the aerosol extinction coefficient above 2 km is usually in the same order of magnitude with the one below 2 km. We estimate that the aerosol extinction coefficient between 2 and 4 km (denote as $AE_{2-4}$) varies from 0% to 100% of the aerosol below 2 km (denote as $AE_{0-2}$), and the median value is 50% of $AE_{0-2}$. In order to estimate the

sensitivity of $O_4$ absorption to $AE_{2-4}$, $O_4$ DSCDs were simulated with profiles with the same aerosol extinction coefficient

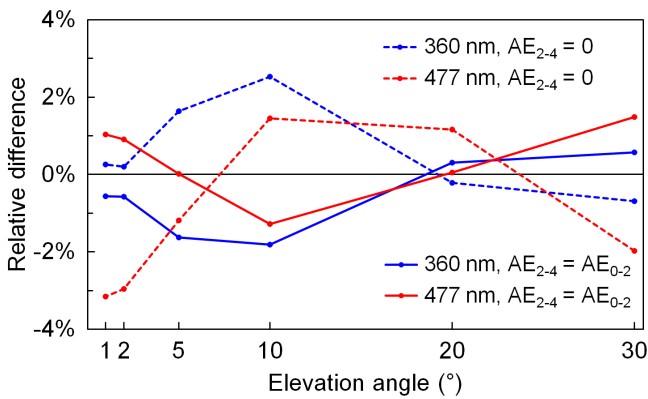

**Figure B5.** Relative differences of $O_4$ DSCDs at 360 nm (blue lines) and 477 nm (red lines) simulated with aerosol profiles with $AE_{2-4}$ (aerosol extinction coefficient between 2 and 4 km above instrument) equals to 0% (dashed lines) and 100% (solid lines) of $AE_{0-2}$ (aerosol extinction coefficient between 0 and 2 km above instrument) comparing to the $O_4$ DSCDs simulated with a profile with $AE_{2-4} = 50\% AE_{0-2}$. For all profiles, $AE_{0-2} = 0.04\,km^{-1}$. The other simulation parameters followed the settings listed in Table B1.

below 2 km ($AE_{0-2} = 0.04\,km^{-1}$) and $AE_{2-4}$ equals to 0%, 50% and 100% of $AE_{0-2}$, and the other parameters were fixed as the settings listed in Table B1. The differences between the $O_4$ DSCDs simulated with extreme and median $AE_{2-4}$ are shown in Fig. B5. The result indicates that the aerosols above 2 km can affect the $O_4$ DSCDs by up to ~3%, which is similar to the surface albedo. Therefore, we consider the influence from the aerosols above 2 km as a kind of measurement uncertainty,

and treat it in the same way as the errors caused by surface albedo, single scattering albedo and phase function uncertainties. Similarly, the uncertainty caused by aerosols above retrieval height is de determined using a simple look-up table in our profile retrieval (see Section 3.7.2).

## B6  Sensitivity to surface aerosol extinction

In order to estimate the sensitivity of $O_4$ DSCD to surface aerosol extinction, $O_4$ DSCDs were simulated with box-shape

profiles with the constant box height of 2 km and aerosol extinction coefficient varies from 0 to $1\,km^{-1}$. The other parameters were fixed as the settings listed in Table B1. The simulated $O_4$ DSCDs are shown in Fig. B6. The result indicates that the sensitivities of $O_4$ absorption at all elevation angles and both wavelength bands decrease with increasing aerosol extinction. Furthermore, the sensitivity is very low when the surface aerosol extinction coefficient exceeds $0.3\,km^{-1}$. The $O_4$ DSCDs at all elevation angles and both wavelengths decrease monotonically with increasing aerosol extinction. In addition, measurements

at lower elevation angles are much more sensitive.

*Competing interests.*  The authors declare that they have no conflict of interest.

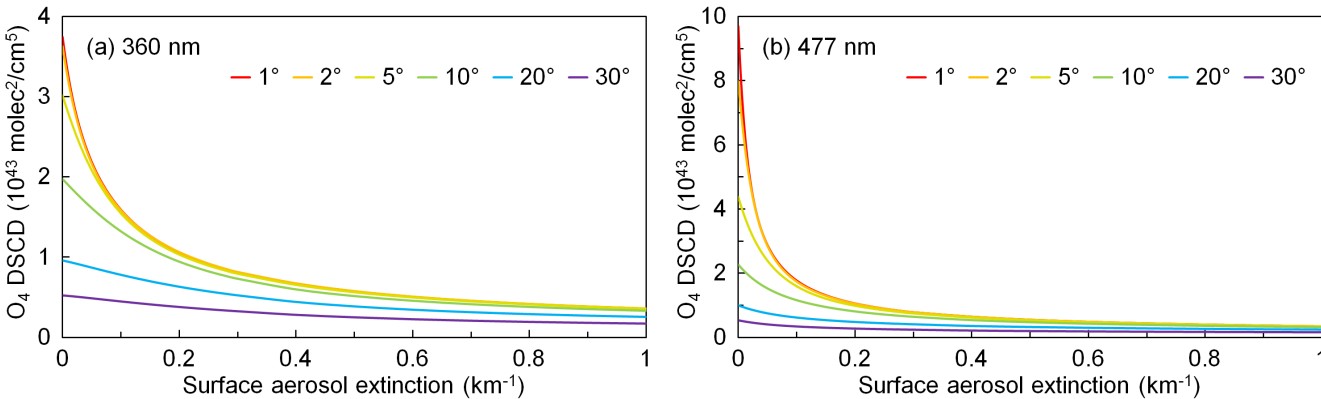

**Figure B6.** Simulated $O_4$ DSCDs at (a) 360 nm and (b) 477 nm for box-shape profiles with the same box height of 2 km and different surface aerosol extinction coefficients from 0 to $1\,\mathrm{km}^{-1}$. The other simulation parameters followed the settings listed in Table B1. Note that the curves of $1°$ and $2°$ are quite close to each other.

*Acknowledgements.* This work is funded by the DLR-DAAD Research Fellowships 2014 (50019750) programme with reference number 91549461. We are thankful for the help from the colleagues of University of Heidelberg in installing the MAX-DOAS instrument at the UFS. We thank Ina Mattis and her colleagues at DWD for their effort in establishing and maintaining the AERONET site in Hohenpeißenberg. The authors would also like to thank the Royal Belgian Institute for Space Aeronomy (BIRA-IASB) for the provision of the QDOAS software used in this study. We are grateful to Till Rehm and his colleagues at the UFS for the maintenance work. We thank the Bayrisches Umweltminsterium for supporting the UFS as part of their mission.

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
