# Peer review of "A MAX-DOAS aerosol profile retrieval algorithm for high altitude measurements: application to measurements at Schneefernerhaus (UFS), Germany"

_Atmospheric Measurement Techniques, 2019_

## Referee Comment (RC1) · Anonymous Referee #1 · 4 Sep 2019

The manuscript entitled "A MAX-DOAS aerosol profile retrieval algorithm for high altitude measurements: application to measurements at Schneefernerhaus (UFS), Germany" by Wang et al. presented a new look-up table based aerosol extinction profile retrieval algorithm for MAX-DOAS observations at Schneefernerhaus (UFS), Germany. Details of the parameterization of the look-up table, retrieval procedure and error analysis are presented. In addition, the authors also investigated the sensitivity of different input parameters to the retrieval results. The new retrieval technique is applied to synthetic data for validation. The long term observations of aerosol optical depth are also

validated by comparing to sun-photometer measurements. The validated MAX-DOAS measurements are then used to investigate the temporal variation of aerosol at UFS. The manuscript is in general well organized and scientifically interesting for the community. Therefore, I recommend publishing the manuscript after addressed the following comments.

Sect.1 para 2: the authors summarize the main methodologies for aerosol monitoring, however, these mentioned AERONET, Lidar and MAX-DOAS are very different in the measured parameters, detection range, etc. I suggest the authors could introduce a little bit about the measured aerosol parameters of these methods, and their advantages and disadvantages for aerosol monitoring.

Sect. 2.2: the sun-photometer measured AOD were interpolated to obtain the AOD at 360 nm and 477 nm. Which kind of the interpolate method? Linear or non-linear? Any large difference due to different interpolate method? Why only time period between 10:00-14:00 UTC and stable aerosol abundance were considered? What does the intensity means in P.5 L.2? The aerosol optical properties required for MAX-DOAS inversion were collected from the AERONET site at Hohenpeißenberg. It is located at an altitude of 980 m and approximately 43 km north of the UFS. As the authors introduced, the aerosol vary strongly with time and location. How to estimate the uncertainties on the retrieved results due to the difference of aerosol optical properties between Hohenpeißenberg and UFS site?

Sect. 3.1: How the DOAS fit windows were determined? Are they based on sensitivity analysis? Please clarify. How about the performance of spectral analysis? The levels of RMS and SCD errors? Any filtering for O4 DSCDs was applied before being introduced to the retrieval scheme? Please add a reference to QDOAS: http://uv-vis.aeronomie.be/software/QDOAS/

Sect. 3.3: How did the authors obtain the topography? And how did the authors distinguish snow or rock and vegetation? Is it taken from a digital elevation map (DEM)

and albedo map? Please clarify. How to define the pseudo-reality topography using TRACY-2? What's kind of the parameters were included in the pseudo-reality topography? It would be useful to compare radiative transfer simulation results from the two radiative transfer models with the same setting to quantify the differences between the two models.

Sect. 3.5: It is difficult to understand the parameterization of aerosol extinction coefficient in Table 3. Please clarify. I also think the vertical resolution of retrieval is very coarse in the design of the look-up table, in particularly compared with other ground-based MAX-DOAS studies. Btw, there only one sub-section of 3.5, I do not suggest to use the title of 3.5.1.

Sect. 3.6: What's the DOAS fitting error? How to evaluate it? There are so many sub-titles. In my opinion, 3.6.1 and 3.6.2 can be grouped as the errors on measured O4 DSCDs, while 3.6.3-3.6.6 can be regarded as the errors on simulated O4 DSCDs. So I suggest to re-organized this part.

Sect. 4.4, p. 26, l. 5-6: Any explanation about the seasonal pattern of AOD that higher in summer and lower in winter? Also the systematic underestimation of MAX-DOAS AOD? Could the authors can present the co-located ceilometer observations or lidar measurements nearby to certificate the vertical structure of aerosol extinction? Please also discuss the possible reason for the high ratio of aerosol extinction coefficient between 360 and 477 nm in summer than in the other seasons.

Sect. 5: The conclusion is mostly repeating the results, please consider shorten the entire summary and conclusion section.

Minor comments: p. 6, l. 2-3: Did the authors observe any seasonal pattern of cloud cover? It might be important for the later analysis of aerosol temporal variation.

p. 7, l. 6: Which radiative transfer model the authors are referring to? Please clarify.

p. 9, l. 11: Please define all the terms in the equation.

[Figure]

p. 13, l. 10: I don't understand why should the surface albedo error dependent on aerosol profile?

p. 14, l. 9-13: If the authors already consider the error caused by aerosol above the retrieval height, then why the error bar of Fig. 9 still do not overlap with the sun-photometer observations most of the time?

p. 17, l. 4-10: Radiative transfer model error also play a role in the discrepancy between measurement and simulation. Please revise the statement. The elevation dependent O4 scaling factor also introduced in other studies, e.g. Irie et al., 2015; Zhang et al., 2019. Please review and cite.

p. 25, fig. 9: The error bars do not overlap with the sun-photometer measurements most of the time indicated that there are some significant error sources are not consider in the error analysis. Please clarify.

p. 27, fig. 11: As mentioned before, cloud screening also play a role in the analysis, it is important to indicate the number of valid measurement used in the calculation.

Reference: Irie, H., Nakayama, T., Shimizu, A., Yamazaki, A., Nagai, T., Uchiyama, A., Zaizen, Y., Kagamitani, S., Matsumi, Y., 2015. Evaluation of MAX-DOAS aerosol retrievals by coincident observations using crds, lidar, and sky radiometer intsukuba, Japan. Atmospheric Measurement Techniques 8, 1013–1054.

Zhang, J., Wang, S., Guo, Y., Zhang, R., Qin, X., Huang, K., Wang, D., Fu, Q., Wang, J., and Zhou, B.: Aerosol vertical profile retrieved from ground-based MAX-DOAS observation and characteristic distribution during wintertime in Shanghai, China, Atmos. Environ., 192, 193–205, 2018.

---

## Referee Comment (RC2) · Anonymous Referee #2 · 4 Sep 2019

**General comments**

Wang et al. introduce a new MAX-DOAS aerosol profiling algorithm for high altitude sites. The algorithm itself is based on a parameterized approach using a pre-calculated look-up table and is optimized to retrieve profiles from data measured at high altitudes. The authors include an extensive sensitivity study and discuss the most important errors thoroughly. Furthermore, an attempt to validate the performance of the algorithm with ancillary measurements is shown. The AMTD version of this manuscript was also

added with one retrieval example of synthetic data and the comparison with an OEM algorithm.

First of all, I would like to comment that the manuscript has improved considerably since the first submission. Unfortunately, my main concern from the first manuscript assessment is still valid. The validation part and the retrieval of synthetic data is not enough to show that this algorithm is not only capable of retrieving accurate profiles but performs better than state-of-the-art OEM algorithms at high altitudes (as the authors claim). In order to solve this issue, I suggest to extend the corresponding sections with the following tests:

1. The retrieval test of synthetic data (Section 4.3) should be complemented with further examples (different exponential profiles and elevated profiles).

2. Comparisons of retrieved profiles with Ceilometer profiles should be added as well. This could be done in an additional section or with similarly averaged Ceilometer profiles in Fig. 10.

3. Fig. 3, 6 and 7 are shown for one example only. It would be interesting to see how the depicted parameters look like for a not so ideal profile retrieval (e.g. smaller (larger) RAA (SZA) or different profile shapes).

**Specific comments**

**P1, L4-5 and P3, L20-23 and P28, L22-23:** The authors claim that commonly used MAX-DOAS algorithms are not suitable for profile retrievals at high altitudes. Since this is neither shown properly in this manuscript nor do the authors cite a publication which addresses this issue, I think that these sentences should be reworded or removed (see also comment to **Section 4.3**).

**P3, L29:** area → areas?

**P4, L15-16:** "The exposure time and number of scans of each measurement are adjusted automatically (...)." Could you please explain what the automatic adjustment of
the number of scans of each measurement means?

**P5, L5-6:** "(...) the derivation of Angström exponents is critical and thus omitted." If this is critical, why is it omitted? In Section 4.5, you derive Angström exponents from MAX-DOAS results. It appears inconsistent to me that Angström exponents are only discussed from MAX-DOAS alone without validating with sun photometer results. This is even more problematic as you found that the MAX-DOAS AODs are much smaller than the sun photometer results (when the AOD lower 0.02 is the main reason for the omission). You could also compare with AERONET data in case the derivation from the available sun photometer is problematic.

**P5, L13:** preformed → performed

**P7, L19-20:** Which phase function and SSA values were used for the simulations? Which climatology was used?

**Section 3.5:** Which climatology was used for the LUT creation? Are different pressure and temperature conditions/profiles are taken into account (in addition to the cross section temperature discussion)?

**P10, L6:** A fixed median phase function was used for the LUT but Fig. 3 and Section B3 tell me that it is quite important to use a proper phase function (especially for small RAA). Do you plan to add more dimensions to the LUT for different phase functions or how do you deal with this problem?

**P11, L22:** ceiometer → ceilometer

**P13, L11-12:** "(...) we found that $O_4$ DSCD at 5° is almost negatively correlated with AOD." This is new for me. Could you please show this in a plot (maybe in the appendix)? Something only valid for high altitude sites?

**Fig. 3:** Please add SZA and RAA values to the description of your example cycle as it is used throughout the manuscript.

**P16, L14-15:** "This is because the a priori profile is not needed in our retrieval algorithm". To be more accurate, you include a priori assumptions of aerosols above retrieval height in your total uncertainty. Furthermore, since your layer $\sigma_3$ and $\sigma_2$ depend on $\sigma_1$ you have another constrain for your solution which could be understood as

a priori information. The question is how do you account for this kind of uncertainty?

**P16, L17:** In $\chi^2 \leq 1.5M$, is M the number of LOS? From an OEM point of view, more LOS mean usually a higher information content. But for your approach, more LOS mean a larger $\chi^2$ criterium and therefore more possible profiles in your weighted mean calculation. Could you please explain this issue?

**Section 4.3:** Please add information on the used OEM algorithm and the RTM including parametrization of the OEM retrieval (e.g the definition of a priori and measurement covariance matrices, climatology, vertical grid...). Maybe in a table? The OEM solutions do not seem to be constrained enough (too many oscillations).

Furthermore, box-like true profiles would also be problematic for lower altitude sites and higher AODs due to the a priori smoothing. Please add also a retrieval for an exponential true profile (an elevated true profile would also be interesting). One problem arises by saying that the shown true profile (nearly box-like) is representative for UFS but you use an exponential a priori profile for the OEM. Since the a priori profile is the best (first) guess of the true atmosphere, an exponential profile is insufficient here (in contrast to typical retrievals in the PBL). A better a priori would be a Boltzman distribution or maybe an exponential profile with an even larger scaling height.

Additionally, please add a graph showing the simulated and retrieved DSCD including an RMS value of the difference between both DSCD as I don't think that the noise-free OEM solutions are that bad but might describe the measurement well.

**Section 4.4 and Fig. 9:** The reason for such a large difference between sun photometer and MAX-DOAS AOD is still unclear to me. I agree that aerosols in higher altitudes might be responsible for a difference but this is true as well for measurements in the PBL. But here, the introduction of a scaling factor leads to a much better agreement. The relative amount of aerosols in altitudes higher than 2km might be responsible but this is just a guess without prove. Could the authors please take a look at the Ceilometer profiles to solve this issue (see also the following comment for Fig. 10)? Furthermore, which kind of scaling factor (SF) is needed to bring the MAX-DOAS AOD to the sun photometer level and how large is the difference to the actually applied

SF? Is there also a clear seasonal pattern in your SF? If yes, maybe your way of how to retrieve LOS depending SF is not optimal?

**Fig. 10:** Please add also similarly averaged Ceilometer profiles and an error range for your profiles. Since a validation instrument is available, a comparison should be shown. The ceilometer backscattering signal could be scaled with the sun photometer AOD (see e.g. Wagner et al. 2019 for details). With this kind of comparison you could also assess how much aerosol is located at even higher altitudes than 2km.

**Section 4.5**: See comment to **P5, L5-6**.

**P29, L6:** profile → profiles

**Appendix B:** Here, important information are missing. For example, when you use aerosols in B1, which SSA and phase function is used? In B3 which SSA? In general, which climatology.

**P34, L8-9:** "using phase functions from Hohenpreißenberg should not have a significant impact on the aerosol retrieval". But the results are only shown for RAA = SZA = $60°$. For other geometries it might be important to have an accurate phase function. Especially since you show in Figure 3 that the phase function is one of the largest error sources.

**P35, L5-6:** the averaging kernels (...) are all close to zero at the altitudes above 2km. That is correct but OEM based aerosol retrievals are iterative approaches which might still get an elevated layer more or less correct even though the kernels look like that. The sensitivity in these altitudes is lower for sure but if there is a dominant elevated aerosol layer, your retrieval is not capable of retrieving it accurately due to the dependencies of the individual layers while OEM algorithms might find an accurate solution (see also comment to **Section 4.3** for a test of an elevated layer).

**References**

Wagner, T., Beirle, S., Benavent, N., Bösch, T., Chan, K. L., Donner, S., Dörner, S., Fayt, C., Frieß, U., García-Nieto, D., Gielen, C., González-Bartolome, D., Gomez, L.,

Hendrick, F., Henzing, B., Jin, J. L., Lampel, J., Ma, J., Mies, K., Navarro, M., Peters, E., Pinardi, G., Puentedura, O., PuÄůÄńte, J., Remmers, J., Richter, A., Saiz-Lopez, A., Shaiganfar, R., Sihler, H., Van Roozendael, M., Wang, Y., Yela, M.: Is a scaling factor required to obtain closure between measured and modelled atmospheric $O_4$ absorptions? An assessment of uncertainties of measurements and radiative transfer simulations for 2 selected days during the MAD-CAT campaign, Atmospheric Measurement Techniques, 2019, https://www.atmos-meas-tech.net/12/2745/2019/, doi:10.5194/amt-12-2745-2019

---

## Author Comment (AC1) · 21 Feb 2020

We thank reviewer #1 for the quick response and the detailed comments. We appreciate the comments and we understand that these comments have a positive effect on the scientific content of the manuscript while improvement of the descriptions and further clarifications are necessary. We have further clarified the retrieval procedure and improved the description of the look-up table parameterization. A point to point basis response to the reviewer's comments is provided in the following for consideration. Our answers are presented in blue texts. Please note that all the page and line numbers

mentioned below refer to the pages and lines in the manuscript with revision marks.

Anonymous Referee #1

The manuscript entitled "A MAX-DOAS aerosol profile retrieval algorithm for high altitude measurements: application to measurements at Schneefernerhaus (UFS), Germany" by Wang et al. presented a new look-up table based aerosol extinction profile retrieval algorithm for MAX-DOAS observations at Schneefernerhaus (UFS), Germany. Details of the parameterization of the look-up table, retrieval procedure and error analysis are presented. In addition, the authors also investigated the sensitivity of different input parameters to the retrieval results. The new retrieval technique is applied to synthetic data for validation. The long term observations of aerosol optical depth are also validated by comparing to sun-photometer measurements. The validated MAX-DOAS measurements are then used to investigate the temporal variation of aerosol at UFS. The manuscript is in general well organized and scientifically interesting for the community. Therefore, I recommend publishing the manuscript after addressed the following comments.

Sect.1 para 2: the authors summarize the main methodologies for aerosol monitoring, however, these mentioned AERONET, Lidar and MAX-DOAS are very different in the measured parameters, detection range, etc. I suggest the authors could introduce a little bit about the measured aerosol parameters of these methods, and their advantages and disadvantages for aerosol monitoring.

Response: We have added brief comments to these measurement instruments in Sect. 1 Para. 2 (see Page 2, Lines 25-30).

Sect. 2.2: the sun-photometer measured AOD were interpolated to obtain the AOD at 360 nm and 477 nm. Which kind of the interpolate method? Linear or non-linear? Any large difference due to different interpolate method?

Response: The data were interpolated following the Angström exponent method. As the AODs at 360 and 477 nm were both interpolated from the data at two nearby wavelengths (340 & 380 nm and 440 & 500 nm, respectively), the difference due to different interpolation method is expected to be small. We have compared our results with the data derived using linear interpolation. For most of the data, the difference is less than 1%. We have supplemented the information about data interpolation in Sect. 2.2 (see Page 5, Line 20).

Why only time period between10:00-14:00 UTC and stable aerosol abundance were considered?

Response: It is because the calibration uncertainty is very high under low SZA. This has been further clarified in the text (see Page 5, Line 21).

What does the intensity means in P.5 L.2?

Response: It means (spectral) radiances. This has been clarified in the text (see Page 5, Line 23). The aerosol optical properties required for MAX-DOAS inversion were collected from the AERONET site at Hohenpeißenberg. It is located at an altitude of 980 m and approximately 43 km north of the UFS. As the authors introduced, the aerosol vary strongly with time and location. How to estimate the uncertainties on the retrieved results due to the difference of aerosol optical properties between Hohenpeißenberg and UFS site?

Response: We realize the different aerosol optical properties between UFS and Hohenpeißenberg. Therefore, we examined the sensitivity of O4 DSCD to aerosol optical properties, and the influence was estimated to be less than 3%. Some other studies also showed that aerosol optical properties only show a small (1.5-4%) impact on the retrieval of aerosol extinction profiles (e.g. Chan et al., 2019). We have supplemented this information to Sect 2.2 Para. 2 (see Page 5, Line 34 to Pate 6, Line 3).

Sect. 3.1: How the DOAS fit windows were determined? Are they based on sensitivity

analysis? Please clarify.

Response: The fit windows were determined according to both the absorption signal of O4 and the SNR of our spectrometers. We have added some explanations in Sect 3.1 (see Page 8, Lines 4-11).

How about the performance of spectral analysis? The levels of RMS and SCD errors? Any filtering for O4 DSCDs was applied before being introduced to the retrieval scheme?

Response: We have added further descriptions about the performance of spectral analysis in Sect 3.1 (see Page 8, Lines 20-23). The results with residual RMS larger than 10-3 are filtered out. This is also supplemented in Sect. 3.1 (see Page 8, Lines 19-20).

Please add a reference to QDOAS: http://uvvis.aeronomie.be/software/QDOAS/

Response: The reference has been supplemented in Sect 3.1 Para. 1 (see Page 7, Line 33).

Sect. 3.3: How did the authors obtain the topography? And how did the authors distinguish snow or rock and vegetation? Is it taken from a digital elevation map (DEM) and albedo map? Please clarify.

Response: Both the altitude data and surface type are obtained from Google Earth. This has been clarified in the caption of Fig. 2 and Sect. 3.3 (see Page 10, Fig. 2 and Line 9).

How to define the pseudo-reality topography usingTRACY-2?

Response: This is described in Sect. 3.3 Para. 4 (see Page 10, Lines 8-14).

What's kind of the parameters were included in the pseudo-reality topography?

Response: These are described in Sect. 3.3 Para. 5 (see Page 10, Line 19 to Page 11, Line 1).

[Figure]

It would be useful to compare radiative transfer simulation results from the two radiative transfer models with the same setting to quantify the differences between the two models.

Response: We have supplemented the comparison result in Sect. 3.3 Para. 3 (see Page 10, Lines 5-7).

Sect. 3.5: It is difficult to understand the parameterization of aerosol extinction coefficient in Table 3. Please clarify.

Response: The description of the parameterization has been refined (see Pages 14-15).

I also think the vertical resolution of retrieval is very coarse in the design of the look-up table, in particularly compared with other ground based MAX-DOAS studies.

Response: The parameterization is based on the measurement sensitivity. Because the information content of the measurements is rather limited, a finer vertical grid would not really improve the accuracy of the retrieval, but would greatly increase the complexity as well as computational effort. In most of the OEM-based MAX-DOAS studies, although their vertical resolution is higher (usually 200 m per layer), the degree of freedom of the signal (DFS) is usually around 2. In addition, the vertical variation of the aerosol extinction at the UFS is expected to be low. Therefore, a coarse resolution setting would be sufficient. This information is supplemented in Sect. 3.5 (see Page 14, Lines 12-15).

Btw, there is only one sub-section of 3.5, I do not suggest to use the title of 3.5.1.

Response: These two sections have been rearranged as Sect. 3.5 and 3.6 (see Pages 14-17).

Sect. 3.6: What's the DOAS fitting error? How to evaluate it? There are so many sub-titles. In my opinion, 3.6.1 and 3.6.2 can be grouped as the errors on measured O4 DSCDs, while 3.6.3-3.6.6 can be regarded as the errors on simulated O4 DSCDs.

So I suggest to re-organized this part.

Response: We have clarified the definition of the DOAS fitting error in the text. This section (now Sect. 3.7) has been rearranged following the reviewer's suggestion (see Pages 17-20).

Sect. 4.4, p. 26, l. 5-6: Any explanation about the seasonal pattern of AOD that higher in summer and lower in winter?

Response: This is explained in Sect 4.5 Para. 1 (see Page 36, Lines 15-16). It can be explained by the higher biogenic emissions in summer, as well as the stronger vertical transport of aerosols.

Also the systematic underestimation of MAX-DOAS AOD? Could the authors can present the co-located ceilometer observations or lidar measurements nearby to certificate the vertical structure of aerosol extinction?

Response: We have supplemented the ceilometer results in Sect. 2.3 (see Page 6, Fig. 1 and Page 7, Lines 13-16). The results indicate that the aerosols above 2 km contribute 30-50% to the total AOD. As the MAX-DOAS reports AOD only up to 2 km, an underestimation of total AOD is expected.

Please also discuss the possible reason for the high ratio of aerosol extinction coefficient between 360 and 477 nm in summer than in the other seasons.

Response: This is explained in Sect. 4.5 Para. 2-3 (see Page 36, Line 21 to Page 38, Line 7). It indicates that the particle size is smaller in summer.

Sect. 5: The conclusion is mostly repeating the results, please consider shorten the entire summary and conclusion section.

Response: We have shortened this section (see Page 38, Line 9 to Page 40, Line 3).

Minor comments: p. 6, l. 2-3: Did the authors observe any seasonal pattern of cloud cover? It might be important for the later analysis of aerosol temporal variation.

Response: We have supplemented a summary of the cloud screening results in Sect. 3.2 (see Page 8, Lines 31-34 and Page 9, Table 2). The percentage of cloudy measurements is highest in summer (67%) and lowest in winter (54%).

p. 7, l. 6: Which radiative transfer model the authors are referring to? Please clarify.

Response: It refers to LIDORT. This has been clarified in the text (see Page 9, Lines 8-9).

p. 9, l. 11: Please define all the terms in the equation.

Response: The missing definition of $\Delta Ss$ has been supplemented (see Page 15, Line 29).

p. 13, l. 10: I don't understand why should the surface albedo error dependent on aerosol profile?

Response: Maybe the previous description was a bit misleading. We have revised the description of Sect. 3.7.2 to avoid confusion (see Page 18, Line 31 to Page 19, Line 4).

p. 14, l. 9-13: If the authors already consider the error caused by aerosol above the retrieval height, then why the error bar of Fig. 9 still do not overlap with the sun-photometer observations most of the time?

Response: This is only a source of error that we consider in the retrieval, but the MAX-DOAS AOD in Fig. 9 only represents the AOD under 2 km. We have further clarified this issue in the text (see Page 17, Lines 3-4).

p. 17, l. 4-10: Radiative transfer model error also play a role in the discrepancy between measurement and simulation. Please revise the statement.

Response: We have revised the description (see Page 24, Line 1).

The elevation dependent O4 scaling factor also introduced in other studies, e.g. Irie et

al., 2015; Zhang et al., 2019. Please review and cite.

Response: We have supplemented the two references (see Page 24, Lines 5-7).

p. 25, fig. 9: The error bars do not overlap with the sun-photometer measurements most of the time indicated that there are some significant error sources are not consider in the error analysis. Please clarify.

Response: The main reason is that the MAX-DOAS AOD only represents the AOD under 2 km. This has been further clarified in the text (see Page 34, Line 33 to Page 35, Line 2).

p. 27, fig. 11: As mentioned before, cloud screening also play a role in the analysis, it is important to indicate the number of valid measurement used in the calculation.

Response: We have supplemented a summary of the cloud screening results in Sect. 3.2. The numbers of valid measurements are listed in Table 2 (see Page 9).

Reference:

Chan, K. L., Wang, Z., Ding, A., Heue, K.-P., Shen, Y., Wang, J., Zhang, F., Shi, Y., Hao, N., and Wenig, M.: MAX-DOAS measurements of tropospheric NO2 and HCHO in Nanjing and a comparison to ozone monitoring instrument observations, Atmospheric Chemistry and Physics, 19, 10 051–10 071, https://doi.org/10.5194/acp-19-10051-2019, https://www.atmos-chem-phys.net/19/10051/2019/, 2019.

Please also note the supplement to this comment:
https://www.atmos-meas-tech-discuss.net/amt-2019-204/amt-2019-204-AC1-supplement.pdf

---

## Author Comment (AC2) · 21 Feb 2020

We would like to thank the reviewers for their valuable comments and suggestions. It took more time than expected to go through all of the individual points, in particular as some of them required a significant extension of our original investigations. E.g., we included additional evaluations of ceilometer data to provide consistency checks for the MAX-DOAS retrievals, as well as additional examples of synthetic data retrieval. As a consequence we want to apologize for the delay of our point to point responses.

[Figure]

Reply to RC2 We thank reviewer #2 for the detailed comments. These comments are useful for use to improve the quality of the manuscript. We have supplemented a more examples of synthetic data retrieval to support our statements. In addition, we have further evaluated the retrieved MAX-DOAS aerosol profile by comparing to ceilometer measurements. We have addressed the reviewers' comments on a point to point basis as below for consideration. Our answers are presented in blue texts. Please note that all the page and line numbers mentioned below refer to the pages and lines in the manuscript with revision marks.

Anonymous Referee #2

General comments

Wang et al. introduce a new MAX-DOAS aerosol profiling algorithm for high altitude sites. The algorithm itself is based on a parameterized approach using a pre-calculated look-up table and is optimized to retrieve profiles from data measured at high altitudes. The authors include an extensive sensitivity study and discuss the most important errors thoroughly. Furthermore, an attempt to validate the performance of the algorithm with ancillary measurements is shown. The AMTD version of this manuscript was also added with one retrieval example of synthetic data and the comparison with an OEM algorithm.

First of all, I would like to comment that the manuscript has improved considerably since the first submission. Unfortunately, my main concern from the first manuscript assessment is still valid. The validation part and the retrieval of synthetic data is not enough to show that this algorithm is not only capable of retrieving accurate profiles but performs better than state-of-the-art OEM algorithms at high altitudes (as the authors claim). In order to solve this issue, I suggest to extend the corresponding sections with the following tests:

[Figure]

1. The retrieval test of synthetic data (Section 4.3) should be complemented with further examples (different exponential profiles and elevated profiles).

Response: We have supplemented two more examples with different profile shapes, one exponential profile, and the other profile with a weak elevated layer. The result shows that our retrieval method can reproduce the true profiles within the measurement error. In addition, the synthetic data were also retrieved by an OEM based algorithm. The results are shown for reference. This result is supplemented in Sect. 4.3 (see Pages 32-34).

2. Comparisons of retrieved profiles with Ceilometer profiles should be added as well. This could be done in an additional section or with similarly averaged Ceilometer profiles in Fig. 10.

Response: We have supplemented the ceilometer results in Sect. 2.3. The ceilometer reports attenuated backscatter, while the MAX-DOAS measures aerosol extinction coefficients. As these parameters are not directly comparable, we have converted ceilometer measurement of attenuated backscatter profiles to aerosol extinction profiles using auxiliary AOD information from the co-located sun photometer. The ceilometer data and the comparison result are shown in Sect. 2.3 (see Pages 6-7).

3. Fig. 3, 6 and 7 are shown for one example only. It would be interesting to see how the depicted parameters look like for a not so ideal profile retrieval (e.g. smaller (larger) RAA (SZA) or different profile shapes).

Response: We have supplemented a few more examples with a smaller SZA and a larger RAA. The new results are shown in Fig. 5 (Page 20), Fig. 9 (Page 29) and Fig. 11 (Page 30).

Specific comments

P1, L4-5 and P3, L20-23 and P28, L22-23: The authors claim that commonly used MAX-DOAS algorithms are not suitable for profile retrievals at high altitudes. Since this

is neither shown properly in this manuscript nor do the authors cite a publication which addresses this issue, I think that these sentences should be reworded or removed (see also comment to Section 4.3).

Response: We have revised the sentences (see Page 1, Lines 5-7, Page 3, Lines 31-34 and Page 38, Lines 14-16). In addition, we have added more examples of synthetic data retrieval. The retrieval of the synthetic data presented in Sect. 4.3 suggested that OEM-based retrievals cannot fully reproduce the true profile. The new results are shown in Sect. 4.3 (see Pages 32-34).

P3, L29: area → areas?

Response: This typo has been corrected (see Page 4, Line 7).

P4, L15-16: "The exposure time and number of scans of each measurement are adjusted automatically (...)." Could you please explain what the automatic adjustment of the number of scans of each measurement means?

Response: We have supplemented the information in Sect. 2.1 (see Page 4, Line 32 to Page 5, Line 4).

P5, L5-6: "(...) the derivation of Angström exponents is critical and thus omitted." If this is critical, why is it omitted? In Section 4.5, you derive Angström exponents from MAX-DOAS results. It appears inconsistent to me that Angström exponents are only discussed from MAX-DOAS alone without validating with sun photometer results. This is even more problematic as you found that the MAX-DOAS AODs are much smaller than the sun photometer results (when the AOD lower 0.02 is the main reason for the omission). You could also compare with AERONET data in case the derivation from the available sun photometer is problematic.

Response: We are sorry for the confusion. As the uncertainty of the AOD measured by the sun photometer is relatively large, the uncertainty of the derived Angström exponent would be further amplified. Consequently, the Angström exponent is not very

reliable, and this is the reason, that they are not discussed in the results. We have supplemented the explanation to Sect. 2.2 (see Page 5, Lines 26-28).

P5, L13: preformed → performed

Response: This typo has been corrected (see Page 8, Line 1).

P7, L19-20: Which phase function and SSA values were used for the simulations? Which climatology was used?

Response: We have supplemented the definitions (see Page 10, Line 20 to Page 11, Line 1).

Section 3.5: Which climatology was used for the LUT creation? Are different pressure and temperature conditions/profiles are taken into account (in addition to the cross section temperature discussion)?

Response: We used the US Standard climatology (Anderson et al., 1986). We have supplemented this information in the text (see Page 17, Table 5). The variation of atmospheric profile (i.e. temperature and pressure) is not considered in the look-up table, while we consider this effect as 'other possible error sources' in the error estimation of the retrieval. We have done a sensitivity analysis with summer and winter atmospheric profiles to estimate the uncertainty related to temperature and pressure variation. We estimated the corresponding uncertainty is less than 2% which is well covered by the 'other possible error sources'. We have supplemented the discussion about this issue in Sect. 3.7.4 (see Page 21, Line 20 to Page 22, Line 3).

P10, L6: A fixed median phase function was used for the LUT but Fig. 3 and Section B3 tell me that it is quite important to use a proper phase function (especially for small RAA). Do you plan to add more dimensions to the LUT for different phase functions or how do you deal with this problem?

Response: Since accurate estimation of phase function is in general not available, it is not feasible to add phase function as an additional dimension. We have further clarified

in Sect. 3.6 that only well-known input parameters are defined as dimensions of the look-up table (see Page 15, Line 30).

P11, L22: ceiometer → ceilometer

Response: This typo has been corrected (see Page 15, Line 11).

P13, L11-12: "(...) we found that O4 DSCD at 5°is almost negatively correlated with AOD." This is new for me. Could you please show this in a plot (maybe in the appendix)? Something only valid for high altitude sites?

Response: Theoretically, aerosol reduces the optical path length for off-zenith measurements in the atmosphere due to enhanced Mie scattering, and the optical path is expected to be the longest under pure Rayleigh atmosphere. Therefore, O4 DSCD is expected to reduce with enhanced aerosol load. Fig. 1 show the correlation between O4 DSCD at 5° and AOD (0-2 km) for all the profiles in the look-up table (SZA = SAA = 60°). In each chart, the trend line is derived by moving average, and the r value is the Pearson correlation coefficient between the original data and the expected values obtained from the trend line. We have revised the description in Sect. 3.7.2 (see Page 18, Line 31 to Page 19, Line 4).

Fig. 3 Please add SZA and RAA values to the description of your example cycle as it is used throughout the manuscript.

Response: We have added SZA and RAA values to the captions of the figures (see Page 20, Fig. 5, Page 21, Fig. 6, Page 25, Fig. 7, Page 26, Fig. 8, Page 29, Figs. 9 and 10, Page 30, Fig. 11 and Page 31, Fig. 12).

P16, L14-15: "This is because the a priori profile is not needed in our retrieval algorithm". To be more accurate, you include a priori assumptions of aerosols above retrieval height in your total uncertainty. Furthermore, since your layer $\sigma 3$ and $\sigma 2$ depend on $\sigma 1$ you have another constrain for your solution which could be understood as a priori information. The question is how do you account for this kind of uncertainty?

Response: Maybe our description was confusing. In the profile set, we excluded profiles with strong elevated layers, but the aerosol extinction in different layers is independent in the retrieval. We have refined the description of the look-up table in Sect. 3.5 to make it easier to be understood (see Page 14, Line 17 to Page 15, Line 21).

P16, L17: In $\chi 2 < 1.5 M$, is M the number of LOS? From an OEM point of view, more LOS mean usually a higher information content. But for your approach, more LOS mean a larger $\chi 2$ criterium and therefore more possible profiles in your weighted mean calculation. Could you please explain this issue?

Response: Yes, M is the number of elevation angles, which is defined in the text (see Page 22, Line 19). According to the definition of $\chi 2$, when M increases, $\chi 2$ would also increase. Therefore, the criterion is not really changed. This is also the case for the stopping criterion of the OEM retrievals. For more details, please refer to Rodger, 2000.

Section 4.3: Please add information on the used OEM algorithm and the RTM including parametrization of the OEM retrieval (e.g the definition of a priori and measurement covariance matrices, climatology, vertical grid...). Maybe in a table? The OEM solutions do not seem to be constrained enough (too many oscillations). Furthermore, box-like true profiles would also be problematic for lower altitude sites and higher AODs due to the a priori smoothing. Please add also a retrieval for an exponential true profile (an elevated true profile would also be interesting). One problem arises by saying that the shown true profile (nearly box-like) is representative for UFS but you use an exponential a priori profile for the OEM. Since the a priori profile is the best (first) guess of the true atmosphere, an exponential profile is insufficient here (in contrast to typical retrievals in the PBL). A better a priori would be a Boltzman distribution or maybe an exponential profile with an even larger scaling height.

Response: We have supplemented two more examples of synthetic data retrieval, one with an exponential profile and one with an elevated layer. The retrieval settings are also supplemented (see Page 34, Lines 5-14). Compared to the previous version, we

set a stronger constraint to the a priori profile for the OEM retrieval. However, further optimization of the OEM retrieval is beyond the scope of this paper.

Additionally, please add a graph showing the simulated and retrieved DSCD including an RMS value of the difference between both DSCD as I don't think that the noise-free OEM solutions are that bad but might describe the measurement well.

Response: The simulated and retrieved DSCDs as well as the RMS of the difference are shown in Fig. 2. Since the manuscript is already very long and the OEM retrieval is not the main focus of the paper, we only briefly summarized the results in the text (see Page 34, Lines 17-18) without showing the plots in the manuscript.

Section 4.4 and Fig. 9: The reason for such a large difference between sun photometer and MAX-DOAS AOD is still unclear to me. I agree that aerosols in higher altitudes might be responsible for a difference but this is true as well for measurements in the PBL. But here, the introduction of a scaling factor leads to a much better agreement. The relative amount of aerosols in altitudes higher than 2km might be responsible but this is just a guess without prove. Could the authors please take a look at the Ceilometer profiles to solve this issue (see also the following comment for Fig.10)?

Response: From the averaged ceilometer profiles we observed that there are significant amounts of aerosols above the MAX-DOAS retrieval height, see Fig. 1. The results indicate that the aerosols above the retrieval height contribute 30-50% to the total AOD. We have supplemented this result in Sect 2.3 (see Page 7, Lines 15-16).

Furthermore, which kind of scaling factor (SF) is needed to bring the MAX-DOAS AOD to the sun photometer level and how large is the difference to the actually applied SF? Is there also a clear seasonal pattern in your SF? If yes, maybe your way of how to retrieve LOS depending SF is not optimal?

Response: As we assume that the scaling factors are constant (for each elevation angle), it is impossible to derive a scaling factor which can bring the MAX-DOAS measurements to the sun photometer observations for all conditions. In addition, since the scaling factors were derived based on a huge amount of data and the AOD varies within a relatively narrow range in a single season, data within a single season are insufficient to derive a representative scaling factor. For example, our determination method for the scaling factors at high elevation angles requires measurements under low aerosol load (AOD<0.03), but such measurements are not available in summer. Therefore, it is not feasible to derive a seasonal pattern of the scaling factors. We have further clarified in Sect. 3.9 (see Page 26, Lines 9-10).

Fig. 10: Please add also similarly averaged Ceilometer profiles and an error range for your profiles. Since a validation instrument is available, a comparison should be shown. The ceilometer backscattering signal could be scaled with the sun photometer AOD (see e.g. Wagner et al. 2019 for details). With this kind of comparison you could also assess how much aerosol is located at even higher altitudes than 2km.

Response: Please note, that the retrieval of the AOD from ceilometer data is per se not possible, at least with a sufficient accuracy to allow a strict validation. To cover the reviewer's point, we can however use ceilometer measurements for a consistency check by considering the paper suggested by the reviewer: we have supplemented the ceilometer profiles scaled using the method described in Wagner et al. 2019 in Fig. 1, and also supplemented error ranges for Fig. 15 (Fig. 10 in the discussion paper). The ceilometer profiles show that the aerosols above retrieval height contribute in the order of 30-50% to the total AOD. This explains the differences between MAX-DOAS and sun photometer AODs. This result was supplemented in Sect 4.4 (see Page 34, Line 32 to Page 35, Line 1).

Section 4.5: See comment to P5, L5-6.

Response: The uncertainty of AOD measured by sun photometer is relatively large, and the deviation of Angström exponent would further amplify the uncertainty. We have clarified in Sect. 2.2 (see Page 5, Lines 26-28).

P29, L6: profile → profiles

Response: This typo has been corrected (see Page 38, Line 22).

Appendix B: Here, important information are missing. For example, when you use aerosols in B1, which SSA and phase function is used? In B3 which SSA? In general, which climatology.

Response: We have supplemented the settings in the text (see Page 41, Lines 11-14 and Page 42, Table B1).

P34, L8-9: "using phase functions from Hohenpreißenberg should not have a significant impact on the aerosol retrieval". But the results are only shown for RAA = SZA = 60°. For other geometries it might be important to have an accurate phase function. Especially since you show in Figure 3 that the phase function is one of the largest error sources.

Response: Fig. B3 only shows an example. In our retrieval, the error caused by phase function is estimated using a look-up table which considered all possible solar and viewing geometries. We have revised the description in Sect. B3 (see Page 45, Lines 12-13).

P35, L5-6: the averaging kernels (...) are all close to zero at the altitudes above 2km. That is correct but OEM based aerosol retrievals are iterative approaches which might still get an elevated layer more or less correct even though the kernels look like that. The sensitivity in these altitudes is lower for sure but if there is a dominant elevated aerosol layer, your retrieval is not capable of retrieving it accurately due to the dependencies of the individual layers while OEM algorithms might find an accurate solution (see also comment to Section 4.3 for a test of an elevated layer).

Response: Our look-up table does not consider extreme cases, i.e. strong elevated layers, as the measurement site is located at a high altitude (2650 m a.s.l.), strong elevated layer is typically either close to our instrument or above the retrieval height.

Strong elevated layers can also be included in the look-up table if it is used for the retrieval of aerosol profiles at low altitude sites. In order to reduce computational efforts, we have limited the formulations of the look-up table with only weak elevated layers. On the other hand, the retrieval of the synthetic data showed that OEM based retrieval cannot fully reproduce the elevated layer, either (see Page 32, Fig. 13). We have revised the descriptions in Sect. 3.5 (see Pages 14-15).

References:

Anderson, G. P., Clough, S. A., Kneizys, F., Chetwynd, J. H., and Shettle, E. P.: AFGL atmospheric constituent profiles (0.120 km), Tech. rep., AIR FORCE GEOPHYSICS LAB HANSCOM AFB MA, 1986.

Chan, K. L., Wang, Z., Ding, A., Heue, K.-P., Shen, Y., Wang, J., Zhang, F., Shi, Y., Hao, N., and Wenig, M.: MAX-DOAS measurements of tropospheric NO2 and HCHO in Nanjing and a comparison to ozone monitoring instrument observations, Atmospheric Chemistry and Physics, 19, 10 051–10 071, https://doi.org/10.5194/acp-19-10051-2019, https://www.atmos-chem-phys.net/19/10051/2019/, 2019.

Rodgers, C. D.: Inverse methods for atmospheric sounding: Theory and practice, vol. 2, World scientific, 2000.

Please also note the supplement to this comment:
https://www.atmos-meas-tech-discuss.net/amt-2019-204/amt-2019-204-AC2-supplement.pdf

[Figure]

[Figure]

**Fig. 1.** Correlation between O4 DSCD at 5° and AOD (0-2 km) for all the profiles in the look-up table (SZA = SAA = 60°).

[Figure]

**Fig. 2.** Simulated and retrieved DSCDs and the RMS for the symthetic study.